# Modeling the Adoption and Diffusion of Mobile Telecommunications Technologies in Iran: A Computational Approach Based on Agent-Based Modeling and Social Network Theory

**Hossein Sabzian, Mohammad Ali Shafia \*, Mehdi Ghazanfari and Ali Bonyadi Naeini**

Department of Progress Engineering, Iran University of Science and Technology, Tehran 16846–13114, Iran; hossein_sabzian@pgre.iust.ac.ir (H.S.); mehdi@iust.ac.ir (M.G.); bonyadi@iust.ac.ir (A.B.N.)

\* Correspondence: omidshafia@iust.ac.ir; Tel.: (+98)9121327677

**Abstract:** Understanding the mechanism underlying the mobile telecommunications technologies (MTTs) diffusion in a country is crucial for telecom planners to know how to accelerate their diffusion by designing appropriate scenarios. Considering the technology diffusion as a bottom-up process, this study is aimed at exploring this mechanism, drawing on insights from diffusion of innovation theory and social network theory. Accordingly, an agent-based model is proposed to investigate how MTTs are diffused in Iran over time. The results of this study show, (1) social network of Iranian society seems more similar to a Watts–Strogatz small-world network than a Barabási–Albert preferential attachment network, where the clustering coefficient is high and average path length is low, (2) compared to the compatibility parameter, the advertisement parameter not only is less influential on diffusion of a targeted MTT (i.e., 4G) but also is not necessary for it, and (3) scenarios having the least number of steps and turning points are more appropriate for continuous diffusion of 4G. The proposed study is empirically validated against real-world data ranging from 7/1/2017 to 12/31/2017. We believe it provides telecom planners insights regarding MTTs diffusion mechanism in a social complex structure and the how of scenario designing for increasing their diffusion.

**Keywords:** Mobile telecommunications technologies (MTTs); Long term evolution (LTE 4G); Diffusion mechanism; Agent-based modeling (ABM); Social network theory (SNT)

## 1. Introduction

High-speed access to mobile data has become one of the most significant demands of smartphone subscribers. The increasing availability of intelligent equipment and the ever-growing demand for multi-media streaming services have drastically increased the volume of mobile data traffic [1]. In 2016, the total generated mobile data was 8.8 Exabyte, 50% of which belonged to video content. It is estimated that this amount will reach 71 Exabyte in 2020, 75% of which will belong to video content [2]. Network operators have spent lots of R&D costs on providing better data. Therefore, it has led to the significant expansion of mobile broadband. Statistics and figures indicate that the mobile broadband subscriber growth rate has been steadily higher than that of fixed broadband in the world. A fixed broadband penetration rate reached 11.9% in 2016, from 0.9% in 2012, while mobile broadband penetration rate reached 49.4% in 2016 from 21.7% in 2012 Figure 1. Like other countries [3], in Iran, mobile broadband penetration rate has been much higher than that of fixed broadband, as the growth rate of fixed broadband subscribers has moved from 2.81% in the summer 2012 only to 12% in the spring 2017, while the growth rate of mobile broadband subscribers has shifted from 1.92% in summer 2012 to 42.5% in spring 2017 Figure 2.

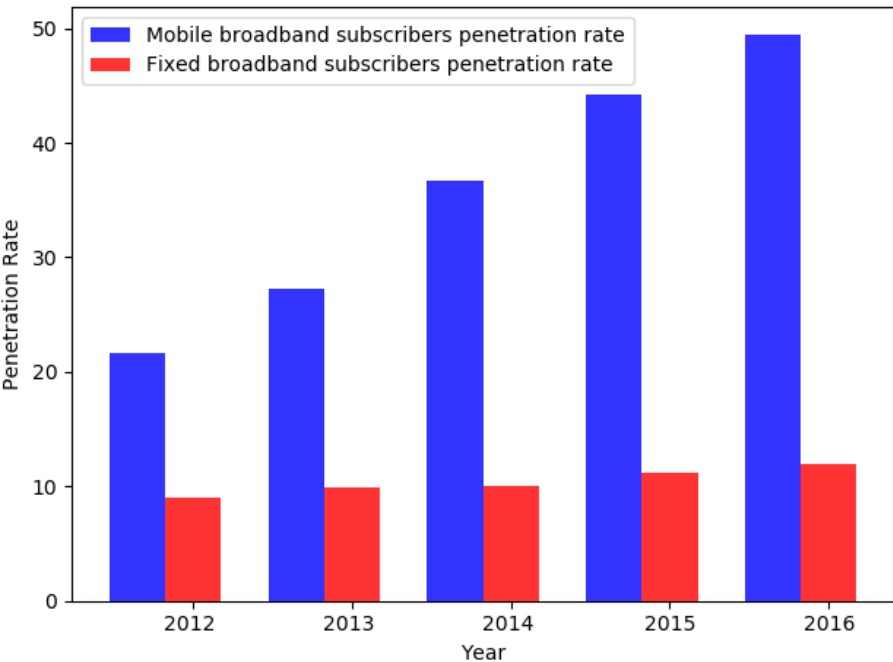

**Figure 1.** Comparison of the penetration rate of mobile broadband subscribers with that of fixed broadband subscribers in the world from 2012–2016 (Source: ITU [4]).

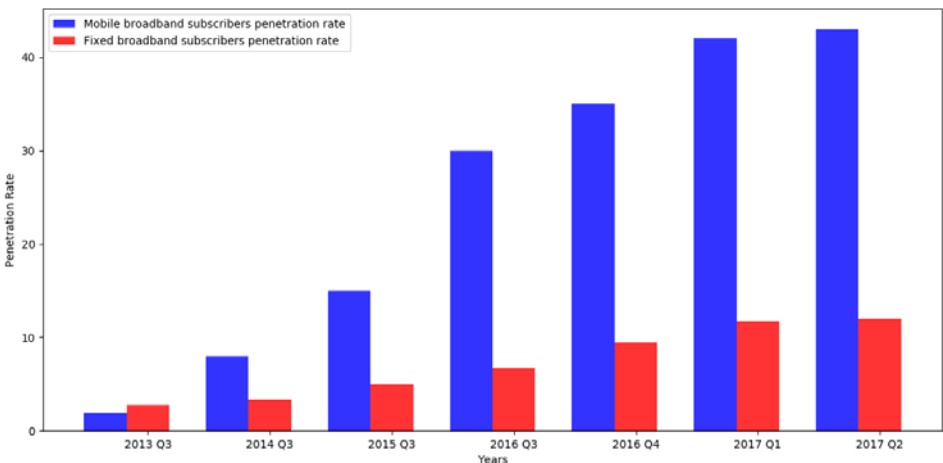

**Figure 2.** Comparison of the penetration rate of mobile broadband subscribers with that of fixed broadband subscribers in Iran from 2013 Q3–2017 Q2 [5].

Cellular network technologies, like the third generation (3G) and fourth-generation (4G) of mobile networks, have considerably contributed to the development of mobile broadband. 3G technology was developed during the 90s and entered the global market in 2002. This technology was becoming widespread in 2007. Actually, in December 2007, 190 operators in 40 countries provided 3G services to their subscribers. In Iran, Rightel operator, which was established in 2007, received the 3G service license in January 2010, but it started its work in June 2011. Therefore, 3G mobile networks entered the market with a delay of 9 years. The commercialization of 3G in the country coincided with the commercialization of 4G in the world. The 4G technology was launched in September 2014 by the Irancell operator. Afterwards, Mobile Telecommunication Company of Iran (MCI) and Rightel started 4G mobile networks. Limitations such as mobility, low-rate of transmission, and bandwidth constraints were among the most important reasons leading mobile operators to go to launch 4G mobile networks.

According to specifications detailed by IMT-advanced, any 4G mobile network should meet the following minimum requirements [6]:

- It should be completely based on Internet Protocol (IPV6)
- For high mobility communications such as from cars and trains, data transmission rate should be 100 megabits per second (Mbps), and for low mobility communications such as pedestrians and stationary users, data transmission rate should be 1 gigabit per second (Gbps)

As the most prominent representative of 4G, Long Term Evolution (LTE) technology has become very popular. LTE was developed in 2004 and commercialized during 2010–2011. LTE, which is often marketed as 4GLTE, is a standard for transmitting high-speed wireless data for data terminals and mobile devices. LTE is based on GSM/EDGE and UMTS/HSPA technologies, which have greatly enhanced the capacity and speed via a different radio interface, together with core network improvements. LTE standard has been developed by the 3rd Generation Partnership Project (3GPP). Differences of mobile telecommunications technologies (MTTs) have been presented in Table 1.

**Table 1.** Different generations of MTTs from 1G to 4G [7].

| Item | 1G | 2G | 3G | 4G |
|---|---|---|---|---|
| Development time | 1970 | 1982 | 1990 | 2004 |
| Commercialization time | 1984 | 1991 | 2002 | 2010–2011 |
| Technical features | Analog voice | Digital voice Short messages | Live podcast speed up to 2Mbps, video form of telecommunication | IP-based, multimedia, long-distance transmission |
| International standard | AMPS, TACS, NMT | GSM, TDMA, DAMPS, CDMA, PDC | WCDMA, CDMA2000, DETC, TDSCDMA | 3GPP |
| Speed | 1.9 Kbps | 9.6–172 Kbps | 382 Kbps-2 Mbps | 86–326 Mbps |
| Application | Voice | Voice and data | Voice, data, video call | voice, data, video call, online game, HD TV |

Several studies have shown the widespread market share of LTE in the next five years so that if mobile operators cannot properly manage this technology, they will eventually lose much of their subscribers' market share. The Ericsson 2016 [2] report has illustrated this trend Figure 3. According to this report, there were 3.9 billion smartphone subscribers in 2016, of which 1700 million are 4G, 1800 million are 3G (UMTS/HSPA), 300 million are 2G (GSM/EDGE), and another one million subscribers use other cellular mobile networks. It is estimated that by the year 2022, 2700 million persons will be added to 4G subscribers and 540 million persons to 5G subscribers, while 3G, 2G, and other mobile network technologies will lose 65 million, 95 million, and 130 million subscribers, respectively.

According to an analysis by the Ministry of Information and Communications Technology (ICT) of Iran [5], the total portfolio of domestic operators was 6250 billion dollars in the year 2015, 74.1% of which was voice and SMS, and only 4.6% was mobile data. With the development of mobile infrastructure and the availability of high-speed broadband, as well as the widespread diffusion of various OTT technologies such as Telegram, a penetration rate of 78%, and Instagram with a penetration rate of 54% by the end of spring 2017, it is forecast that of the 20,937,500,000 dollar revenues for domestic operators in 2020, only 35.1% would be voice and SMS, and 26.9% for mobile data. It reflects the fact that by 2022, the share of voice and SMS will be halved, and the data share will roughly get six times higher. The portfolio revenue change of Iranian operators is shown in Figure 4.

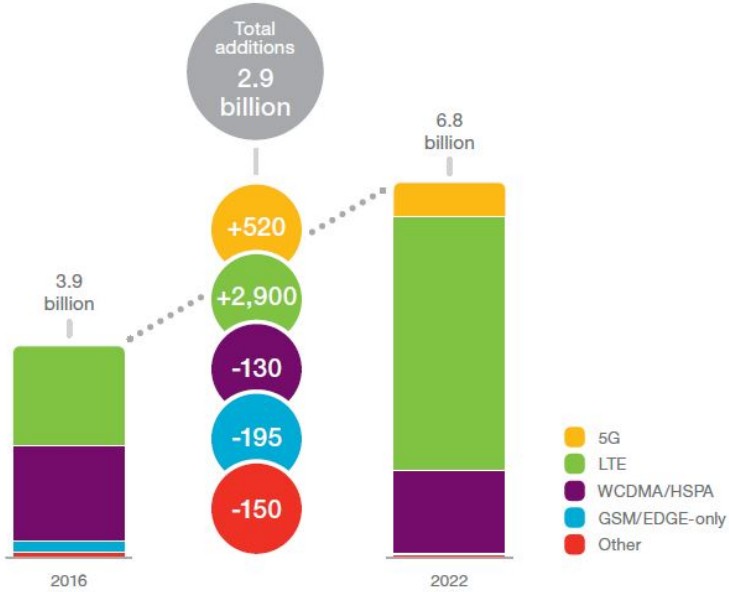

**Figure 3.** Number of smartphone users according to the type of cellular technology [2].

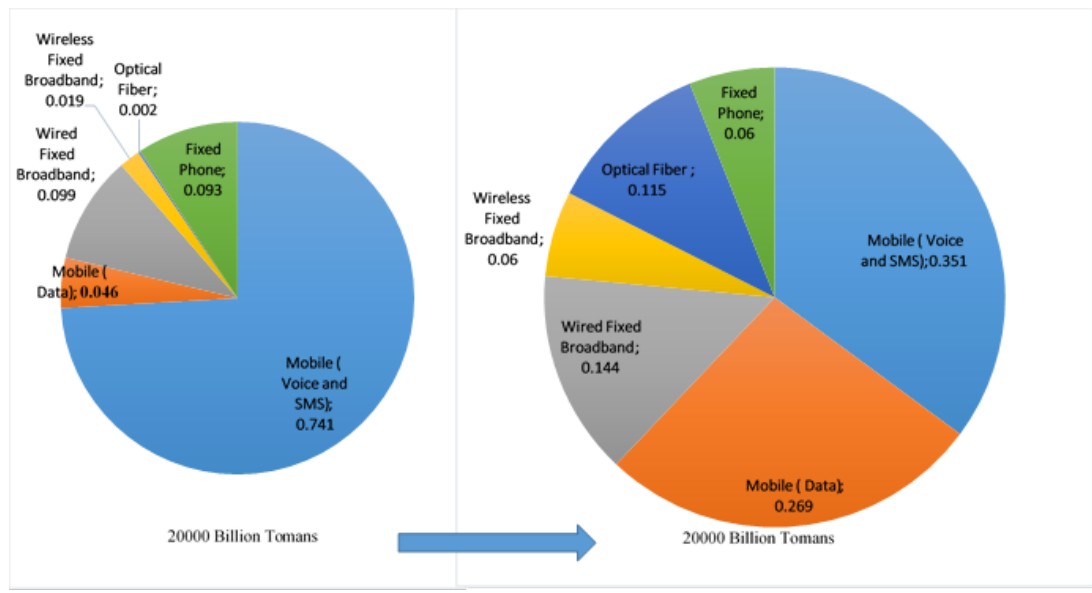

**Figure 4.** Evolution of revenue portfolio of Iranian Operators from 2015–2020 [5].

Since the introduction of LTE technology for the first time in Iran (i.e., September 2014), domestic mobile operators have been investing heavily in developing this technology. The increase in coverage of this network from 47.86% in November 2016 to 58.82% in June 2017 indicates an increase in the number of 4G transmitter/receiver base stations (BTS) across the country. However, in spite of the large investments of the three main domestic operators, namely MCI, Irancell, and Rightel, in the development of 4G LTE network, as well as the increasing penetration of mobile broadband in the country—about 42.5% by the end of the spring of 2017—the LTE mobile penetration rate in Iran is estimated to be 10% by the end of spring 2017. This indicates that the main volume of mobile broadband subscribers in the country is still using UMTS/HSPA 3G networks. In other words, about 26 million of total mobile broadband subscribers (i.e., 34 million subscribers by the end of spring 2017) are currently using 3G cellular networks. Given the mobile broadband penetration rate and the third-generation share in it (85%), it is clear that mobile broadband subscribers are locked in these

third-generation UMTS/HSPA networks, and the poor reception of the fourth generation will slow down future investments in LTE networks. This can result in some unintended negative consequences as following:

- *Failure to provide a large number of high-quality services:* With the help of LTE technology, operators can provide much more diverse and high-quality services. Some of the major advantages of this technology include faster mobile data, more bandwidth, much less latency, better management of data traffic, high-level video streaming, Voice over LTE (VoLTE), Video over LTE (ViLTE), entry to vertical markets (e.g., public safety, health and transportation), low legacy cost due to more integrability with fifth-generation networks (5G), network's better efficiency, cell broadcast, and lots of value-added services.
- *Inability to compete with OTT technologies:* Because LTE operators can provide better quality services than OTT technologies, such as VoLTE versus Voice over Internet Protocol (VOIP) offered by WhatsApp or ViLTE compared to the video over the Internet protocol provided by Skype.
- *Loss of many revenue opportunities:* The revenue generated by LTE is forecast to be $ 350 billion by 2020 [8]. Considering that Iran is 1.08% of the world's population, its share of this amount can be equal to 3.780 billion. So, if the fourth generation is not well understood, Iranian operators will lose that income share.
- *Prolongation of payback period:* If operators' subscribers migrate slowly from 3G to 4G, this can prolong the payback period of the 4G network and create many costs for operators.
- *The lack of integrability with fifth-generation networks (5G) and the loss of many IoT opportunities:* IoT is recognized to have a tremendous effect over the ICT industry in the near future. The Forbes Media Company forecasts that the IoT revenue share will be $ 14.4 trillion by 2022. Of this, $ 2.5 trillion has been spent on improving employees' productivity, another 2.5 trillion in reducing costs, 2.7 trillion in improving logistics and supply chains, 3 trillion in reducing market entry time, and 3.7 trillion in improving the customers' experience [9]. 5G plays a pivotal role in realizing the IoT and utilizing its capacities. 4G technologies, in particular, LTE-Advanced Pro technology, have a high potential to integrate with 5G in comparison to other cellular technologies. With this technology, operators can get integrated with 5G networks without getting stuck at a huge legacy cost. Therefore, if the operators only stay on 3G networks, huge costs will be imposed on them to migrate to 5G networks, and they will lose huge opportunities.

Therefore, it is important for managers of Iranian operators to know by which scenarios they can encourage non-4G subscribers to switch to fourth-generation technologies. For such scenarios, they should know the mechanism underlying the diffusion of three competing MTTs (i.e., 2G, 3G, and 4G) in the country. This study is primarily aimed at discovering this mechanism. The rest of this paper is organized as follows: The second section covers the literature review where various types of studies are discussed, and their limitations are explained. The third section ideals with materials and methods where agent-based modeling (ABM) is discussed in a step-by-step way. In the fourth section, the results are presented through model evaluation in which both verification and verification are clearly discussed. In the fifth section, a discussion of the model's outputs is presented, and finally a conclusion is elaborated as the sixth section.

## 2. Literature Review

In our complex world, we observe a variety of complex phenomena such as new social norms formation, rumor spreading, the emergence of new technologies, and the fall of traditional businesses. To study phenomena, social sciences researchers most often use reductionism approach, through which they reduce such phenomena to some lower-level variables and model the relationships among them through a scheme of equations (e.g., partial differential equations and ordinary differential equations). This reductionism approach, which is often called equation-based modeling (EBM) has a long background in the field of mobile technology adoption, where most of these EBM studies can

be classified into two classes of (1) statistical regression-based studies [10–14] and (2) Multi-Criteria Decision Making (MCDM) ones [15–20]. Statistical regression-based models have some basic limitations. The first limitation of statistical regression-based models is that they are formulated based on some simplistic assumptions that can distort the accuracy of solutions. For instance, in linear regression, the variables are considered to be linearly related in which the independent variables can take a range of values, while dependent ones only have a random value. Additionally, successive observations of independent variables are regarded to be uncorrelated, and the co-linearity among variables has to be taken into consideration [21]. The second limitation is that such models are very prone to the prediction error; therefore, there can be a great gap between the outcome variable and real-world data. The third limitation is that in multiple regression, the beta value is indirectly calculated by a test, so in case of a negative beta, the final judgment becomes really difficult [18]. MCDM studies have been frequently used to identify the major criteria affecting the adoption of mobile services. Liu (2010) used the Analytical Hierarchy Process (AHP) to prioritize the critical factors of mobile handset usage on the basis of employees' opinions. In that study, the critical factors were divided into four factors of (1) economic value, (2) relational value, (3) knowledge value, and (4) convenience value, that the convenience value was finally selected as the most critical factor [15]. Phan and Diam (2011) ranked and clustered the major influencers on mobile service adoption using AHP and cluster analysis. Their study included five total factors of habits, social factors, technology, ease of use, and usefulness. The results showed that the last two factors play a vital role in the adoption process [16]. Regardless of business aspects, Nikou and Mezei (2013) could rank the factors impacting students to adopt mobile services using an AHP approach. They identified 20 mobile services in five distinct categories. The results of their study indicated that five services of "short message," "mobile email," "mobile internet surfing," "mobile search," and "mobile Google map" were believed to be the most important services affecting mobile adoption [17]. However, none of these studies dealt with the diffusion problem in a crisp manner, so the lack of a fuzzy assessment is their biggest limitation.

Buyukozkan (2009) developed a fuzzy AHP (FAHP) to prioritize the requirements of mobile experiences according to organizational criteria and opinions expressed by users and experts. However, considering a small number of criteria is considered to be the prime limitation of this study [19]. Combining the FAHP and extent analysis approach, Lin (2013) developed a fuzzy assessment model for prioritizing factors affecting the quality of mobile banking. The opinions of two different groups were used in this study, one of which possessed a little experience of mobile banking, while another one had a high level of experience in it. However, the results show that two groups had chosen the criterion of "customer services" as the most critical factor influencing the effectiveness of electronic banking [20]. Sheih and colleagues (2014) used FAHP to rank critical factors of mobile services adoption by Taiwan citizens. They grouped all factors into three groups of "factors related to mobile," "factors related to mobile hardware," and "psychological factors of users" [18]. Nevertheless, these studies had two major shortcomings. The first of which was that the cause and effect relationships among factors affecting technology adoption were not considered, and the second one was that the intensity of these relationships was not also taken into consideration.

Generally, as it can be observed in all EBM studies of technology adoption and diffusion, in modeling how a technology is adopted and diffused in a society over time, the whole society is reduced into some socio-economic factors with the qualities of unbounded rationality and (most often) perfect information, and the model built from the relationships among such factors is analyzed to explain the technology adoption and diffusion in the society, while very important factors such as adaptability and the evolutionary nature of all engaged actors along with network effects go unaddressed.

For overcoming deficiencies and limitations of reductionism approach, in the past two decades, the framework of complex adaptive system (CAS) has shown lots of promise. In contrast to reductionism approach, under this framework, the socio-economic phenomena such as technology adoption and diffusion are studied in an organic manner where the economic agents are supposed to be both boundedly rational and adaptive. Based on CAS framework, the socio-economic aggregates such as

technology diffusion emerge out of the ways agents of a socio-economic system interact and decide. As the most powerful methodology for CAS modeling, agent-based modeling (ABM) has gained an increasing application among academia and industry. ABMs show how simple behavioral rules of agents and local interactions among them at a micro-scale can create surprisingly complex patterns at a macro-scale. This methodology has gained an increasing popularity, both in academia and industry such as the study of critical infrastructures dependence [22], consumption of services of sustainable eco-systems [23], economic sciences [24,25], and management [26,27]. For adoption and diffusion studies, a number of works have used ABM for studying the adoption and diffusion of technologies such as environmental technologies [28], energy technologies [29], health and medical technologies [30–32], agricultural technologies [33], cinema technologies [34], transportation technologies [35], electricity production [36], alternative fuels [37,38], organic farming practices [39], automobile markets [40], fuel cell vehicles [41,42], mobile phone [43], and smart metering [44].

However, to the best of authors' knowledge, no systematic study has been conducted on the adoption and diffusion of mobile telecommunications technologies (MTTs) so far. Using agent ABM and social network theory (SNT), the prime purpose of this study is to discover the mechanism underlying the diffusion of three competing MTTs in Iran, through which all telecoms planners would get practical insights in order to speed up the diffusion of a targeted MTT.

## 3. Materials and Methods

### 3.1. Agent-Based Modeling

Agent-based modeling (ABM) is a type of computational model that explores systems of multiple interacting agents which are spatially situated and can evolve over time. ABMs are very effective in exhibiting how complex patterns emerge from micro-level rules over a time period. In contrast to equation-based models (EBMs), such as statistical models and partial differential equations [45] that are rooted in deductive inference, ABMs can not only work as an inductive inference technique where a conclusion is drawn from a series of observations, but also as a pure form of abductive inference by which the best explanation for the phenomena under the study is captured by simulation. ABMs have become a significant modeling trend in a large number of domains ranging from epidemics studies [46] and bio-warfare threat [47] to formation of social norms [48], optimization of supply chain [49], and cooperation in project teams [50].

Unlike EBMs that basically focus on the relationship among macro-variables of a system in a top-down manner, using a bottom-up approach, ABMs try to model how local and non-trivial interactions among micro-level elements of a system are able to generate a complex macro-level behavior of that system [51]. ABM methodology is grounded on complexity theory and network science. In terms of complexity theory, ABMs are developed to explain how simple rules generate complex emergence, and in terms of network science, ABMs are applied to analyze the pattern that arises from agents' interactions over time [52]. ABMs involve three building blocks of (1) agents, (2) environment, and (3) interactions [52–54]. Agents as the first building block of ABMs are the basic computational units of agent-based models. They are characterized by two main aspects of (1) properties and (2) behaviors (or rules of action). The agent's properties are internal or external states that can be changed by its behaviors, and the agent's behavior is a set of rules based on which it decides to act. As the second building block of ABMs, the environment entails all conditions all around the agents as they interact within the model. In other words, the environment is where an artificial social life unfolds [54]. As the third building block of ABMs, interactions refer to rules of behaviors for both agents and the environment [54]. These rules actually enable agents to interact with both themselves and others. All building blocks of ABM of this study will be discussed in the following.

### 3.2. Development of ABM

Generally, an ABM can be developed through three successive and often iterative phases (1) Designing, (2) implementation, and at last, (3) evaluation and output analysis [52]. In the designing phase, the initial skeleton of ABMs is made. This primary skeleton is actually the textual model of ABM into which all behavioral rules and properties of agents and environment, along with the way they interact with each other, are verbally presented and documented. As the second phase, implementation phase (or programming phase) deals with how to translate the ABM's textual model to a computational model by agent-based programming languages and toolkits. The evaluation and output analysis phase as the third phase is conducted for getting insights concerning (1) model verification, (2) model validation, and (3) model's output analysis.

### 3.2.1. Model Designing

Through the designing phase, all rules of behavior and properties of agents, topological environment, and the way agents interact with each other are documented by the natural language of modeler(s). This documentation actually serves as a textual model (this textual model is the conceptual model of an ABM, which is documented in natural language).

ABMs designing can be viewed from two aspects of (1) *modeling purpose* and (2) *model development approach*.

In terms of modeling purpose, ABMs can be divided into two main categories of *phenomena-based modeling* and *exploratory modeling* [52]. In case of phenomena-based modeling, researchers start with a known targeted phenomenon. Typically, that phenomenon has a specific pattern, known as a reference pattern. In exploratory modeling, a researcher can create a set of agents, define their behavioral rules, and explore the patterns that emerge. One might explore them merely as abstract forms similar to the case of cellular automata developed by Conway [55].

In terms of model development approach, all ways used for developing ABMs range from *theory-based modeling* to *evidence-based modeling* [56]. ABMs can be developed via a theory or a set of theories, actually a theory that determines the behavioral rules of agents or the statistical regularities that the model is designed to explain them. In contrast to theory-based ABMs that are constructed upon prior empirical studies and individual-level data [52,57–60]. ABMs can also be developed on the basis of evidence (i.e., personal perceptions and intuition). The evidence-based ABM is applied when researchers have a mental model concerning behavioral rules of agents of a system, and they are interested in understanding the collective behavior of that system when its agents interact with each other [61–63]. The relationship between modeling purpose and modeling development is Figure 5.

In terms of modeling purpose, the model of this study is phenomena-based, because it is designed to simulate the diffusion of three MTTs over a specified time period, and in terms of model development approach, the model of this paper is both theory-based and evidence-based. It is mainly theory-based where real-world empirical data, theories, and extension of the works of [64–67] have been taken into account (Every so often, previously built models are reused or extended by other researchers. This strategy, which is called TAPAS "Take A Previous model and Add Something," is widely used in the ABM community. In this strategy, modelers take an existing model and successively modify it through adding new features or relating initial assumptions [68]), and it is (a little) evidence-based because some new variables (e.g., handset compatibility) are added by researchers. The model is discussed through nine following stages.

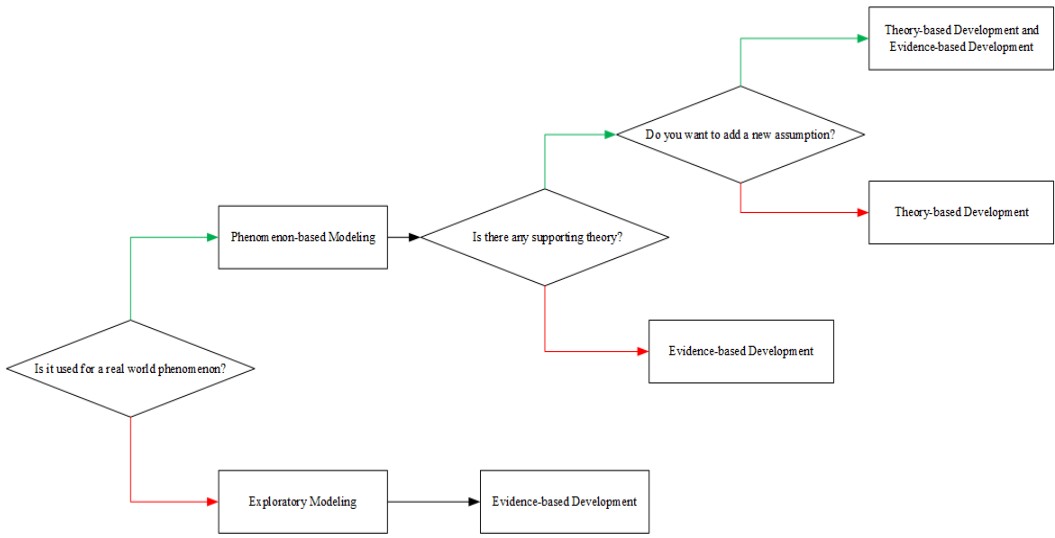

**Figure 5.** The relationship between modeling purpose and modeling development.

Underlying Question of the Model

The proposed model has been developed to help telecom strategists better answer the following questions:

*The major question: What is the mechanism underlying the diffusion of three competing MTTs (i.e., 2G, 3G, and 4G) in Iran?*
*The first sub-question: What are the elements of this mechanism?*
*The second sub-question: How are these elements interrelated?*
*The third sub-question: Which social complex network is more similar to that of Iran?*
*The fourth sub-question: What are the critical drivers of MTTs diffusion in the country?*
*The fifth sub-question: What are the appropriate scenarios for accelerating 4G technology diffusion?*

Types and Classes of Agents

Type of agents refers to agent breeds used in the ABM [52], and an agent class includes several agents sharing a similar behavioral rule(s) [58]. In this model, human-type agents have been used in five classes of (1) innovators, (2) early-adopters, (3) early-majority, (4) the late majority, and (5) laggards. These classes are based on diffusion of innovation (DOI) theory of Rogers (2003), that is going to seek to explain why, when, how, and at what rate innovations spread. DOI is a generally accepted model among technology diffusion researchers from various disciplines such as health and clinical studies [69,70], information systems [71], electronic devices [72], mobile computing and software technologies [73,74], and social media diffusion [75]. For a comprehensive study of the applications of DOI, look at [76].

Rogers defines the diffusion as a process through which an innovation is communicated over time among the actors of a social system. According to him, an adopter category is defined as a categorization of individuals within a social system on the ground of innovativeness degree, which is designated by reluctance to innovation (R2I) score in this paper. He proposes a total of five categories of adopters including innovators, early adopters, early majority, late majority, and laggards.

All these categories were extracted using individual innovativeness (II) scale developed by Hurt and colleagues [77], which can been immediately accessed at http://www.jamescmccroskey.com/measures/innovation.htm. The II Scale was distributed among a sample of randomly selected Iranian citizens (i.e., 1250 individuals, out of which 903 returned, and 800 of them were complete), and the statistical results of R2I scores of respondents matched the categorization of DOI as visualized in Figure 6.

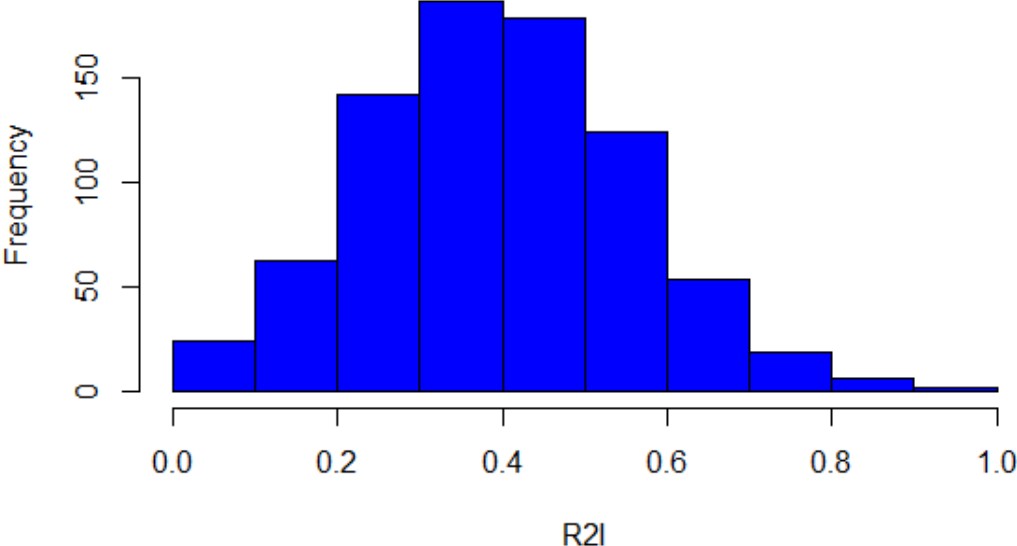

**Figure 6.** Statistical distribution of Reluctance to Innovation (R2I).

Innovators are eager to take risks and are willing to test the unknown. This category forms 2.5% of adopters and has a very low degree of reluctance to innovation (R2I), ranging from 0 to 0.088. They are often young and social, have financial liquidity, and closest contact to scientific communities and other innovators. Their tolerance of risk (i.e., low degree of R2I) enables them to use technologies that are likely to fail; at the end, their financial resourcefulness sufficiently allows them to absorb such failures.

For early adopters, these individuals have the highest degree of opinion leadership among the adopter categories, making up 13.5% of adopters. Early adopters possess a higher social status, advanced education, high financial liquidity, and are more socially oriented than late adopters. These people are more discreet in testing the unknown (i.e., adoption choices) than innovators. Individuals of this category often try to maintain a central communication position. The R2I of this category ranges from 0.088 to 0.233. For the class of early majority, members of this class adopt an innovation after a time period that is much longer than two classes of the innovators and early adopters. This category forms 34% of adopters, with a R2I value ranging from 0.233 to 0.3912. Members of this category class have social status, which is regarded to be above average. They communicate well with early adopters, and rarely take positions of opinion leadership in a system.

For the late majority, they adopt an innovation after the average participant. This category forms 34% of adopters. They are used to approaching with a high level of doubt. Actually, they switch to a new technology after it is adopted by the majority of society. The social status of members of this category is below-average and they possess a little financial liquidity and a low degree opinion leadership. The R2I of this category ranges from 0.3912 to 0.5565. For laggards, they are the last ones to adopt an innovation. This category forms 16% of adopters. These individuals typically show an aversion to change-agents such as innovators or early adopters. In contrast to previous categories, adopters in this category show little to no opinion leadership. Laggard adopters are majorly focused on "traditions," and oldest among adopters. Likewise, they have lowest social status, lowest degree of financial liquidity, and are only in contact with family and close friends. The R2I of this category ranges from 0.5565 to 1.

Granularity of Agents

ABMs can bear various agents with different granularities, ranging from micro-scale to macro-scale [52]. All agents used in this model are of human-scale granularity in five classes of (1) innovator, (2) early adopter, (3) early majority, (4) late majority, and (5) laggard.

Properties of Agents

An agent's properties are internal variables of the agent which change over time. Properties are essential in shaping the behavioral rules of agents. Properties of agents of this study are detailed in Table 2, where they are described and their designated symbols and scales are presented. In addition, development methods and data source of these properties are presented in Table 3.

**Table 2.** Properties of agents.

| Property | Symbol | Description | Scale |
|---|---|---|---|
| Two-G? | $2G\text{-}Agent_i$ | Is $agent_i$ using 2G mobile technology? | Logical |
| Three-G? | $3G\text{-}Agent_i$ | Is $agent_i$ using 3G mobile technology? | Logical |
| Four-G? | $4G\text{-}Agent_i$ | Is $agent_i$ using 4G mobile technology? | Logical |
| Innovator? | $IN\text{-}Agent_i$ | Is $agent_i$ an Innovator? | Logical |
| Early-adopter? | $EA\text{-}Agent_i$ | Is $agent_i$ an early adopter? | Logical |
| Early-majority? | $EM\text{-}Agent_i$ | Is $agent_i$ an early majority? | Logical |
| Late-majority? | $LM\text{-}Agent_i$ | Is $agent_i$ a late majority? | Logical |
| Laggard? | $L\text{-}Agent_i$ | Is $agent_i$ a laggard? | Logical |
| 4G-Compatible? | $4GComp\text{-}Agent_i$ | Is the handset device of $agent_i$ compatible with the 4G network? | Logical |
| Reluctance to Innovation | $R2I\text{-}Agent_i$ | The extent to which $agent_i$ feels reluctant to adopt new technologies. The more it is, the more slowly agent acts to adopt new technology. | Ratio |
| Technology lifetime | $TL\text{-}Agent_i$ | The maximum time that an agent supposes to use a technology which is computed by ($R2I\text{-}Agent_i$ * D), where D is a constant value. | Ratio |
| Technology age | $TA\text{-}Agent_i$ | It is set to 0 when that technology is adopted and advances by one unit in every tick of time. | Ratio |
| Change threshold | $CT\text{-}Agent_i$ | It is computed by rounding ($R2I\text{-}Agent_i$ * $N\text{-}Agent_i$). | Ratio |
| 2G-specific memory | $2GM\text{-}Agent_i$ | For $agent_i$, it saves both the number of its friends that have adopted 2G technology and itself, if it is using 2G. | Ratio |
| 3G-specific memory | $3GM\text{-}Agent_i$ | For $agent_i$, it saves both numbers of its friends that use 3G technology and itself, if it is using 3G. | Ratio |
| 4G-specific memory | $4GM\text{-}Agent_i$ | For $agent_i$, it saves both the number of its friends that have adopted 4G technology and itself, if it is using 4G. | Ratio |

Note: * stands for multiplication.

The development source column refers to how the properties or parameters have been developed. Since this model is both theory-based and evidence-based, some properties and parameters are directly taken from theories [64,65] while some other parameters are new and inserted by researchers to address the question of interest (Evidence). Item of change column refers to the point that how much authors have changed theory-based properties. A property may be renamed and its computing rule may be changed (Name and Computing rule) by authors, as is the case of technology lifetime ($TL\text{-}Agent_i$). This property was named "car-lifetime" in one of the underlying models [64], and was computed according to a random normal distribution function while the authors changed its name and computing rule. Additionally, there may be some properties that have only been renamed (Name), as is the case of 2G-specific-memory ($2GM\text{-}Agent_i$), which was named "diesel list." The data source column deals with the ways data are extracted for setting values (i.e., calibrating) of the properties. In this table, the individual innovativeness (II) scale for measuring reluctance to innovation (R2I) score of respondents and recorded social data (RSD) have been used for setting values of properties. II scale is an standard instrument designed by Hurt and colleagues [77], which was used for measuring R2I of a sample of Iranian citizens about which there was no existing data. The statistical distribution of R2I, which has been extracted via analyzing data gathered from II scale, is presented in Tables 4 and 5. RSD was

provided by Iran Telecommunication Research Center (ITRC) as it is presented in provider column. This organization is the most experienced research entity of ICT filed in the country.

**Table 3.** Development and data source of properties.

| Symbol | Development | Item of Change | Data Source | Provider |
|---|---|---|---|---|
| 2G -Agent$_i$ | Theory [64] | Name | RSD | ITRC |
| 3G -Agent$_i$ | Theory [64] | Name | RSD | ITRC |
| 4G -Agent$_i$ | Evidence | - | RSD | ITRC |
| R2I-Agent$_i$ | Theory [64,65] | - | II Scale | - |
| IN-Agent$_i$ | Theory [64,65] | - | R2I | - |
| EA-Agent$_i$ | Theory [64,65] | - | R2I | - |
| EM-Agent$_i$ | Theory [64,65] | - | R2I | - |
| LM-Agent$_i$ | Theory [64,65] | - | R2I | - |
| L-Agent$_i$ | Theory [64,65] | - | R2I | - |
| 4GComp- Agent$_i$ | Evidence | - | RSD | ITRC |
| TL-Agent$_i$ | Theory [64] | Name, Rule | - | - |
| TA-Agent$_i$ | Theory [64] | Name | - | - |
| CT-Agent$_i$ | Theory [64] | - | - | - |
| 2GM-Agent$_i$ | Theory [64] | Name | - | - |
| 3GM-Agent$_i$ | Theory [64] | Name | - | - |
| 4GM-Agent$_i$ | Theory [64] | Name | - | - |

**Table 4.** Input parameters.

| Parameter | Symbol | Description | Scale |
|---|---|---|---|
| Number of agents | N | All agents | Ratio |
| 2G adopters | N$_{2G}$ | Percentage of all agents as number of 2G adopters | Ratio |
| 3G adopters | N$_{3G}$ | Percentage of all agents as number of 3G adopters | Ratio |
| 4G adopters | N$_{4G}$ | Percentage of all agents as number of 4G adopters | Ratio |
| Advertisement | Ad | Advertisement volume of the latest technology | Ratio |
| 4G-compatibility | 4GComP | The percentage of the population (2G and 3G users) that has a 4G-compatible handset device | Ratio |
| The memory capacity of innovator-type | MC$_I$ | Memory capacity limitation of innovator-types | Ratio |
| The memory capacity of early adopter-type | MC$_{EA}$ | Memory capacity limitation of early-adopter-types | Ratio |
| The memory capacity of the early majority-type | MC$_{EM}$ | Memory capacity limitation of the early majority-types | Ratio |
| The memory capacity of the late majority-type | MC$_{LM}$ | Memory capacity limitation of the late majoritytypes | Ratio |
| The memory capacity of laggard-type | MC$_L$ | Memory capacity limitation of laggard-types | Ratio |
| Network | PA Network | Preferential attachment network | Nominal |
| | WSN | Watts-Strogatz network | Nominal |

**Table 5.** Development and data source of Input parameters.

| Symbol | Development | Item of Change | Data Source | Provider |
|---|---|---|---|---|
| Number of agents | Theory [64] | Name | RSD | ITRC |
| 2G adopters | Theory [64] | Name | RSD | ITRC |
| 3G adopters | Theory [64] | Name | RSD | ITRC |
| 4G adopters | Evidence | - | RSD | ITRC |
| Advertisement | Evidence | - | RSD | ITRC |
| 4G-compatibility | Evidence | - | RSD | ITRC |
| The memory capacity of innovator-type | Theory [64] | Name | EV | - |
| The memory capacity of early adopter-type | Theory [64] | Name | EV | - |
| The memory capacity of the early majority-type | Theory [64] | Name | EV | - |
| The memory capacity of the late majority-type | Theory [64] | Name | EV | - |
| The memory capacity of laggard-type | Theory [64] | Name | EV | - |
| Network | Theory for PAN [66] and WSN [67] | - | - | - |

Input Parameters

Input parameters are independent variables that researchers want to see their influence on the model's behavior. A parameter can have different scales and ranges, out of which combinations or scenarios can be made. Each of these scenarios can have different effects on the system in multiple runs (This is only because of the random structure of ABMs [78]). All input parameters used in the model are presented in Tables 4 and 5.

In the data source column of Table 5, an EV means the data has been directly extracted from empirical values (EV) of prior empirical studies [64,65].

Environment

An environment is all conditions surrounding the agents when they interact within the model. In other words, the environment is where an artificial social life unfolds [54]. Environments can be divided into three different forms of (1) *spatial* (2) *networked,* and (3) *mixed*. The spatial environment is most often a discrete environment, entailing several discrete points. A lattice structure is regarded as the most common form of a spatial environment. In real-world settings, such as socio-economic situations, agents have more networked interactions than spatial interactions. For networked interactions, a number of network structures have been proposed, and two of them are widely used for social structures, which are "Watts–Strogatz small-world network" (WSN) [67] and "scale-free preferential attachment network" (PAN) [66]. These networks are visualized in Figure 7. Using network structures as an ABM environment provides lots of opportunities to synthesize social network theory (SNT) with ABM. ABMs are developed to explain how simple rules generate complex emergence, and in terms of network science, ABMs are used to analyze the pattern that arises from agents' interactions over time [52]. When the spatial environment and networked environment come together, they form a mixed environment. In this paper, two social network structures of WSN and PAN have been used.

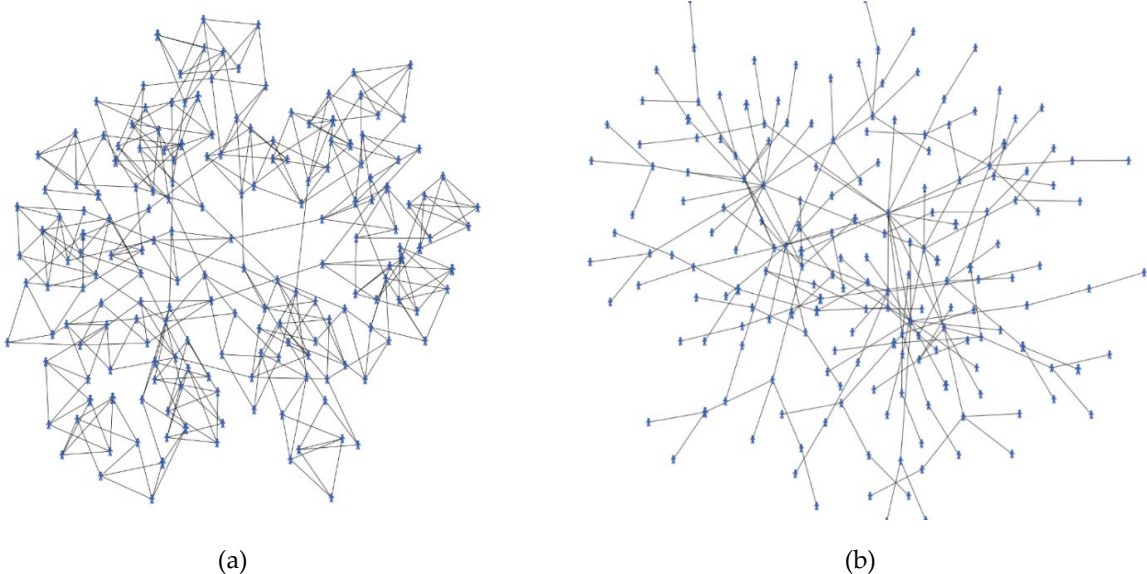

(a)                                                        (b)

**Figure 7.** A small word network (**a**) and a preferentail attachment newtork (**b**).

Small world networks and scale-free networks are considered as two prominent network topologies that are frequently observed in real-world phenomena. A small-world network is a kind of mathematical graph where most of nodes are not neighbors of one another, but the neighbors of any given node are likely to be neighbors of each other, and most nodes can be reached from every other node by a small number of steps (Real-world social networks such as cooperation networks, power grids, friendship networks, and so on show the property of the small-world effect. Such an effect is commonly observed

in human society. It most of times happens that when two persons meet each other, they very soon find out that they have a common friend in between, therefore they both say: "What a small world!" A very interesting example of the "small-world effect" is concept of "six degrees of separation" principle, proposed by a social psychologist, Milgram, in the late 1960s [79]). A Watts–Strogatz small-world network (WSN) stands between regular lattice networks [80] and ER random graphs [81]. A WSN not only shows a small average path length (L), but also a high clustering coefficient (CC). In the analysis of social networks, Average path length is a concept in network topology that is defined as the average number of steps along the shortest paths for all possible pairs of network nodes., and clustering coefficient (CC) refers to the extent to which one's friends are also friends [79]. In network science, clustering can be both local and global. The local clustering coefficient of a node is used to measure how connected its neighbors are. It is represented as the number of links between the node's neighbors divided by the total number of possible links between its neighbors. The global clustering coefficient is used to measure how many nodes tend to cluster together in the entire network. It is defined according to the types of triplets in the network. A triplet is composed of a central node and two of its neighbors. If its neighbors are connected as well, it is a triangle (a closed triplet). If its neighbors are not connected, it is called an open triplet. The global clustering coefficient is the number of triangles in a network divided by the total number of triplets (both open and closed). The L value of a WSN grows proportionally to the logarithm of the number of nodes N in the network [67], and the CC value of a WSN tends to $\frac{3}{4}$ in case of large N [82].The degree distribution of a small-world network obeys the familiar Poisson distribution, and the shape of the Poisson distribution falls off exponentially [67].

While other types of networks, such as regular lattice, Erdős–Rényi (ER) random graph model, and the Watts–Strogatz (WSN) model do not show power laws, approximately a number of real-world networks fall into the class of scale-free networks, meaning that they possess power-law degree distributions. In comparison with small-world networks, this power-law distribution falls off more gradually than an exponential one, allowing for a few nodes of the very high degree to exist (i.e., hubs). Because these power-laws are independent of any characteristic scale, such a network with a power-law degree distribution is called a scale-free network. In terms of average path length, a scale-free network resembles to a small-world one, but is a lower clustering coefficient [80]. The Barabási–Albert model, known as preferential attachment network, is one of several suggested models that create scale-free networks. This model integrates two important general concepts of: Growth and preferential attachment, which widely exist in real networks. Growth signifies that the number of nodes in the network intend to increase over time. Preferential attachment means that the more links a node have, the more likely it is to get newer links. Higher-degree nodes have a higher ability to get links added to the network [66]. In this paper, both of these network structures have been separately used to simulate the social environment where adopters interact and a technology is diffused.

Behavioral Roles of Agents

Agents take action based on their behavioral rules. Behavioral rules represent the mental model according to which agents decide and then act. The rules are a function of input parameters and agent properties, and can range from very simple to very complex [26]. In Figure 8, a flowchart has been drawn to visualize the behavioral rules of the model. All symbols are extracted from properties (Table 2), and input parameters (Table 4), and the k symbols refer to a continuous random number generated in the open interval of (0,1). In this figure, black-colored lines indicate "Yes," and red-colored ones indicate "No."

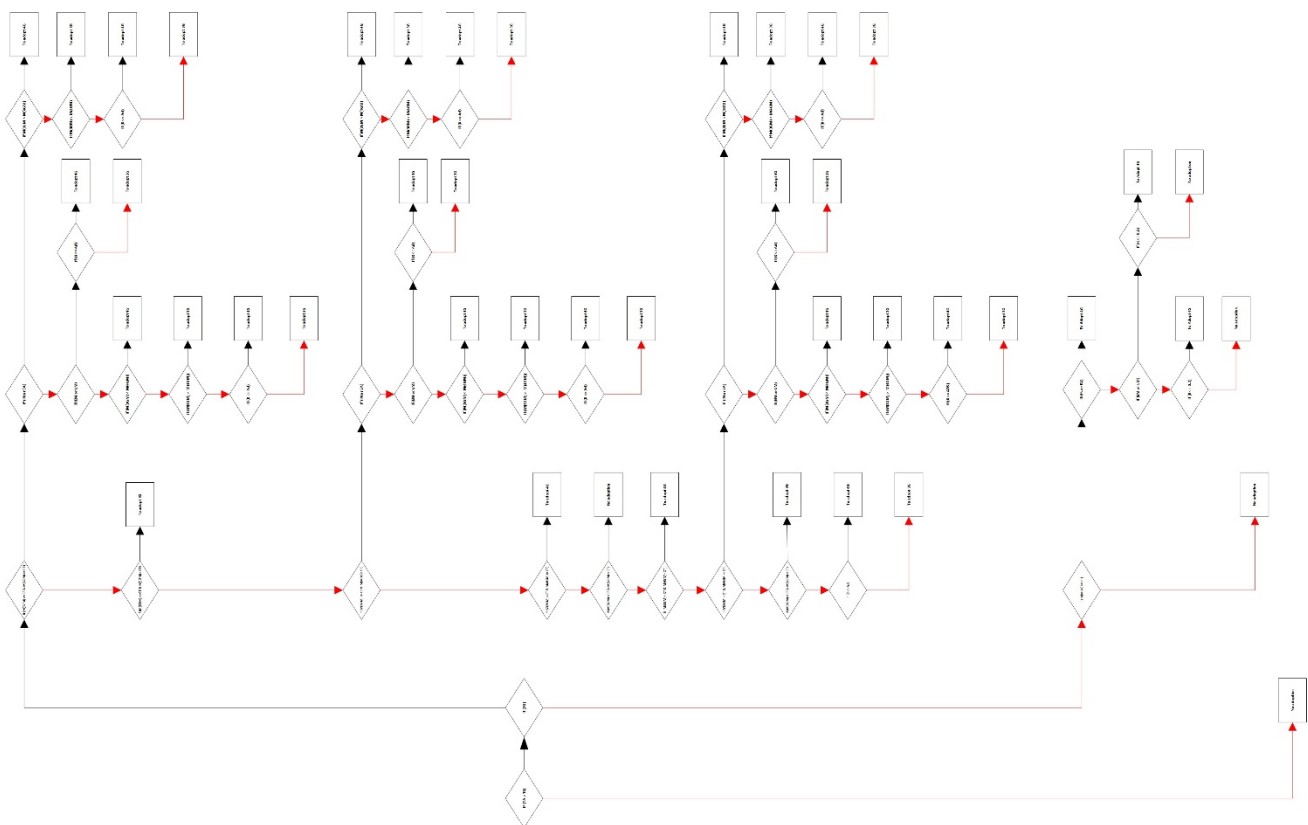

**Figure 8.** Flowchart of behavioral rules of agents.

Sequence of Behaviors

The bottom-up structure of the flowchart shows all evaluations (as diamonds) and actions (as rectangles). In this structure, all the above layers are dependent on the below ones.

Output Measures

Three MTTs of 2G, 3G, and 4G (LTE) are the outputs of this model that are simultaneously simulated over a specified time period ranging from the beginning of summer 2017 to the end of autumn 2017.

### 3.3. Model Implementation Software

After designing an ABM, agent-based programming languages or simulation toolkits have to be used to simulate it. These simulation toolkits are a type of simulation software specially made for translating the textual model of an ABM into a computational model. A simulation is an understandable manifestation of a model, coded and visualized by a computer program, which provides insights regarding the system under study. A simulation model basically refers to the computing algorithms or mathematical expression that involve the performance and total behavior of a system in the real-world scenarios [83]. In the early 1990s, general-purpose programming languages (GPPLs) were basically used for simulation. SMALLTALK, C++, and Java were the most common GPPLs in the ABM community of practice. Using GPPLs for agent-based simulation has some visible disadvantages. For example, modelers have to implement basic functions and plotting from scratch, and they should be very familiar with the programming language [84]. Such problems have resulted in the development of agent-based simulation toolkits that help modelers a lot to simulate the complex system under study. As presented in Table 6, the majority of toolkits support the primary GPPLs, including Java, C++, C, and Logo variant [85]. It should be noted that sometimes, an ABM can be directly developed in the programming phase through participatory simulation platforms. This type of simulation is useful for simulating systems that there is not enough data about; therefore, designing an initial conceptual model for them is almost impossible. According to this approach, an agent-based model is directly developed through direct participation of stakeholders of the problems in distributed platforms such as the client-server network. This simulation is very useful in research and education [86,87].

**Table 6.** Toolkit and their programming languages.

| Toolkit | Programming Languages | | | |
| --- | --- | --- | --- | --- |
| | Java | C++ | C | Logo |
| ADK | * | | | |
| AgentBuilder | * | * | * | |
| AnyLogic | * | | | |
| Ascape | * | | | |
| DeX | | * | | |
| Echo | | | * | |
| iGen | * | * | * | |
| LSD | | * | | |
| Madkit | * | * | * | |
| MAML | | | * | |
| Mason | * | | | |
| Netlogo | | | | * |
| RepastS | * | | | |
| Starlogo | | | | * |
| StarlogoT | | | | * |
| Swarm | * | | | |

Note: * stands for support.

As it can be seen in the above table, some of the platforms are supported by multiple languages such as AgentBuilder, iGen, and Madkits that are supported by Java, C++, and C. During this paper, Nelogo 6.0.1 has been used to simulate the model (see the user interface in Appendix A). As one of the most frequently used agent-based modeling and simulation toolkit, Netlogo was developed by Uri Wilensky in 1999. Since its development, it has been regularly updated in the sequence of versions and a number of extensions. Readers can refer to Netlogo home page ccl.northwestern.edu/netlogo/ in order to get more information about this agent-based modeling toolkit.

## 4. Results Evaluation

An implemented model can never be considered scientific unless its results are evaluated in terms of verification and validation. Without verification and validation, a model is more a toy than a tool [88].

### 4.1. Model Verification

In the verification process, the primary purpose is to determine whether the designed (conceptual) model corresponds to the programmed (implemented) model. As it is sensible, through a verification process, the researchers try to understand the gap between the designed model and its implementation (computational model) and fill it through correction and code debugging. In fact, the verification process comes into play when the model author and model programmer (implementer) are different persons, which is very common among academic researchers cooperating in a team [89,90]. In such a situation, the model authors should iteratively discuss the model with the model programmer so that the model programmer understands and programs what exactly model authors want. This process is called *"iterative modeling"* [52], which can help model verification process when the model author and model programmer are different persons. One very noteworthy point for facilitating the verification process is that the model authors frequently check the textual model developed in the designing phase and find its inconsistencies with the computational model developed in programming. The model of this study was verified through an iterative discussion between model authors and the programmer, though the programmer himself was one of the authors. The verification process, executed through iterative modeling, is very similar to code walkthroughs practice, in which model authors examine a code and the model implementer walks them through each step, explaining what the code is expected to do [88].

### 4.2. Model Validation

When a model is implemented, its validation becomes so essential. A valid model assures the researchers of the model's rightness [91]. Therefore, the model's results are supposed to be useful out of the model, and can be confidently used for policymaking. Validation is generally defined as when the simulated model produces the results that are in a satisfactory range of accuracy, matching up with the real-world data [57]. Validation of computational models has always been a significant concern for simulation specialists [57,90,92,93].

In the big picture, all validation methods can be classified as qualitative and quantitative methods. Qualitative methods, like face validity, are very subjective and expert-based, while quantitative ones, such as empirical validation approaches, are objective and based on real-world data [91]. Face validation is the process of showing that the mechanisms and properties of the model look like mechanisms and properties of the real world. It is mainly conducted by subject matter experts (SMEs). Empirical validation assures that the model generates data that can correspond to similar patterns of the real-world data. Selection of validation approaches to ABMs primarily depend on modeling purpose [56]. When ABMs are used to simulate a real-life statistical regularity, as is the case of phenomena-based modeling, empirical validation approaches are often used. Nevertheless, when ABMs are applied to simulate the mental models of system stakeholders and show what will emerge out of them, as is the case of exploratory-based modeling, qualitative validation methods such as face validity are often applied. In addition to type of validation approach, since ABM is a two-level model

(i.e., individual agent vs. system's global behavior), it has to be validated both at micro level and macro level. The relationship between validation level and validation type has been presented in Table 7.

**Table 7.** The relationship between validation type and validation level.

| Type \ Level | Face | Empirical |
|---|---|---|
| Micro | Face micro validation | Empirical micro validation |
| Macro | Face macro validation | Empirical macro validation |

In face micro validation, SMEs analyze how much the properties, input parameters, and behavioral rules of agents are in face similar to the reality. In face macro validation, SMEs analyze how much model's outputs are in face similar to the reality. In the empirical micro validations, researchers want to know how much the model's input parameters, properties, and behavioral rules correspond to real-world data. This type of validation is often referred to as direct calibration [94] or input empirical validation [90]. In empirical macro validation, the prime purpose is to understand how much the outputs of an implemented model correspond to the real-world data. Since the model of this study is phenomena-based trying to simulate the diffusion of three MTTs, so the empirical validation is used at two levels.

### 4.2.1. Empirical Micro Validation

Empirical micro validation typically deals with calibrating model's input parameters according to empirical data. The empirical data used for setting the values of input parameters of this model are drawn from recorded social data (RSD) provided by ITRC and empirical values (EV) of one of the underlying studies [64]. These empirically derived values are presented in Table 8.

**Table 8.** Empirically derived parameters.

| Parameter Symbol | Development | Item of Change | Data Source | Provider | Derived Value |
|---|---|---|---|---|---|
| N | Theory [64] | Name | RSD | ITRC | 800 |
| $N_{2G}$ | Theory [64] | Name | RSD | ITRC | 440 |
| $N_{3G}$ | Theory [64] | Name | RSD | ITRC | 280 |
| $N_{4G}$ | Evidence | - | RSD | ITRC | 80 |
| Ad | Evidence | - | RSD | ITRC | 0.6 |
| 4GComP | Evidence | - | RSD | ITRC | 0.8 |
| $MC_I$ | Theory [64] | Name | EV | | 0.1 |
| $MC_{EA}$ | Theory [64] | Name | EV | | 2 |
| $MC_{EM}$ | Theory [64] | Name | EV | | 4 |
| $MC_{LM}$ | Theory [64] | Name | EV | | 6 |
| $MC_L$ | Theory [64] | Name | EV | | 8 |
| PAN | Theory [66] | - | - | | TRUE or FALSE |
| SWN | Theory [67] | - | - | | TRUE or FALSE |

As we alluded to above, there are many different ways to set (i.e., calibrate) the parameter values of an ABM. In our case, all RSD values were provided by ITRC. Empirical values (EV) of $MC_I$, $MC_{EA}$, $MC_{EM}$, $MC_{LM}$, and $MC_L$ were directly drawn from [64], which in turn was calibrated from earlier data. In the case of network inputs, we used a different approach; because we were less sure of what inputs to use, we analyzed two variants of networks [66,67] that have been reported to be representative of real-world social networks, and investigate the sensitivity of model's performance in each of two variants. Now the model's inputs are empirically validated and realized; the *empirical macro-validation* should be implemented.

### 4.2.2. Empirical Macro Validation

Empirical macro validation of the model's output includes showing that the output of the implemented model corresponds to the real world [88]. This is a key test for the model's validity. Empirical output validation tests the model designer's hypothesis (i.e., the implemented model). There are three different ways to empirically validate a model's output: *Real-world data, stylized facts, and cross-validation*.

Real-world data validation is crucial if the model is going to be used as a predictive model. In this case, there may often be only one real historical data set, and many runs of the model; thus, empirical output validation includes showing that the real world is a possible output of this model (i.e., that the real-world dataset lies within the statistical distribution of the model data). If there are many outputs from the model and the real world, then real-world data validation includes showing that the average model results correlate with the real-world results [57,94]. A stylized fact is a general concept of a complex system derived from the knowledge of domain experts. If the model's major purpose is to be a used as a thought experiment, then validating the model against a stylized fact is enough to show the validity of the model [90]. Cross-validation (or docking) is an optional validation process that compares the new model against another model that has already been validated, even if that other model uses another methodology (e.g., EBMs such as system dynamics). If the two models generate similar results, then the validity of the new model has been enhanced [92].

In this study, the outputs of the model were empirically validated against real-world datasets ranging from the beginning of summer 2017 to the end of autumn 2017 (i.e., six months). Since the model is a multi-output model, three various datasets (2G, 3G, and 4G) have been used for its output validation. These datasets were provided by ITRC and are visualized in Appendix B. Empirical validation of a model's outputs includes three parts of (1) number of experiments, (2) data normality testing, and (3) correlation coefficient measurement. In case of empirical outputs validation, it should be noted that if its results don't show an acceptable level of accuracy in matching up with real-word data, the model' elements, such as properties, input parameters, and even behavioral rules, have to be re-calibrated. The process of model's output empirical validation is visualized as Figure 9.

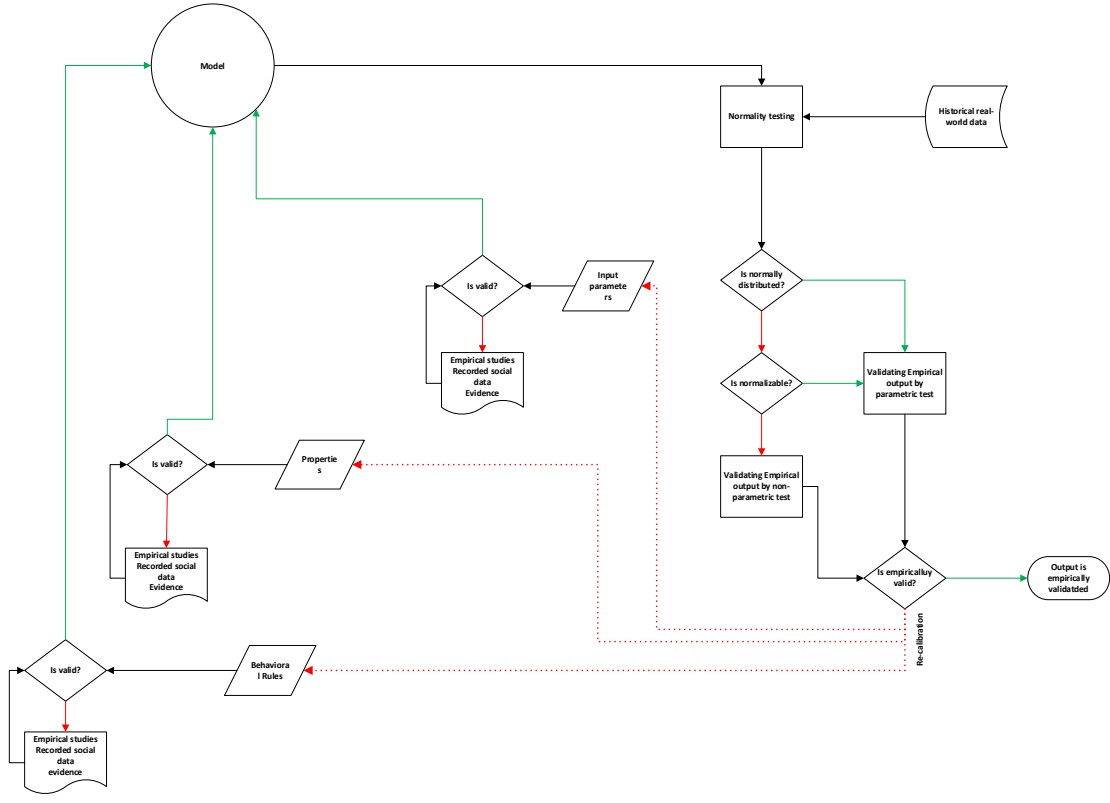

**Figure 9.** Steps of output empirical validation and its impact on model's elements re-calibration.

Number of Experiments

Since ABMs have a stochastic nature, they produce various outputs, even if they have one specified combination of input parameters. So, the statistical distribution of the model's multiple experiments (i.e., runs) would be considered essential in determining its correspondence with real-world data [52,78,90].

The presented model in this study is a three-output model where the diffusion of three competing telecoms technologies (i.e., 2G, 3G, and 4G) is simultaneously simulated while there were two mutually exclusive social networks in which the combination of numeric values (As provided in Table 8) should be tested. For this purpose, using *BehaviorSpace* tool, the model was randomly run by 1500 experiments for each network type, which resulted in 1500 experiments for the performance of every MTT in each of social networks (i.e., totally 900 experiments with positive random seeds ranging from 1 to 1500) as presented in Table 9. This has majorly helped better understand which type of social network fits that of our case study.

**Table 9.** Model's outputs experiments.

| | | MTTs | | |
| | | 2G | 3G | 4G |
|---|---|---|---|---|
| **Network Structures** | **PAN** | Experiment 1<br>.<br>.<br>.<br>Experiment 750<br>.<br>.<br>.<br>Experiment 1500 | Experiment 1<br>.<br>.<br>.<br>Experiment 750<br>.<br>.<br>.<br>Experiment 1500 | Experiment 1<br>.<br>.<br>.<br>Experiment 750<br>.<br>.<br>.<br>Experiment 1500 |
| | **WSN** | Experiment 1<br>.<br>.<br>.<br>Experiment 750<br>.<br>.<br>.<br>Experiment 1500 | Experiment 1<br>.<br>.<br>.<br>Experiment 750<br>.<br>.<br>.<br>Experiment 1500 | Experiment 1<br>.<br>.<br>.<br>Experiment 750<br>.<br>.<br>.<br>Experiment 1500 |

Data Normality Testing

To get a deeper insight concerning the predictive accuracy of the model's outputs, the correlation coefficient test should be implemented. To find that which correlation test has to be used, knowing the normality of the data (both model's outputs and real-world data) is necessary. So, using R statistical programming language, a Shapiro-Wilk test [90] was performed for deciding the normality of all real data (i.e., 2G, 3G and 4G) and simulation data. As it is presented in Table 10, since each of model's outputs had 1500 experiments (i.e., runs), each of them was divided into fifteen 100-run blocks. Every block shows a probability distribution resulting from the average of 100 different experiments of each telecoms technology in each network structure type (i.e., each telecoms technology's average simulated data in 100 runs in each of network structures).

**Table 10.** Division of model's outputs experiments into 15 blocks.

| | | MTTs | | |
| | | 2G | 3G | 4G |
|---|---|---|---|---|
| **Network Structure** | **PAN** | Block 1 (average of 1 – 100 experiments)<br>Block 2 (average of 101 – 200 experiments)<br>.<br>.<br>.<br>Block 15 (average of 1401 – 1500 experiments) | Block 1 (average of 1 – 100 experiments)<br>Block 2 (average of 101 – 200 experiments)<br>.<br>.<br>.<br>Block 15 (average of 1401 – 1500 experiments) | Block 1 (average of 1 – 100 experiments)<br>Block 2 (average of 101 – 200 experiments)<br>.<br>.<br>.<br>Block 15 (average of 1401 – 1500 experiments) |
| | **WSN** | Block 1 (average of 1 – 100 experiments)<br>Block 2 (average of 101 – 200 experiments)<br>.<br>.<br>.<br>Block 15 (average of 1401 – 1500 experiments) | Block 1 (average of 1 – 100 experiments)<br>Block 2 (average of 101 – 200 experiments)<br>.<br>.<br>.<br>Block 15 (average of 1401 – 1500 experiments) | Block 1 (average of 1 – 100 experiments)<br>Block 2 (average of 101 – 200 experiments)<br>.<br>.<br>.<br>Block 15 (average of 1401 – 1500 experiments) |

In Table 11, $H_0$ indicates the data is normally distributed, column 3 of this table shows the results of Shapiro–Wilk test of data, and as observable for all cases, the p-value is far less than 0.05. Therefore, $H_0$ is rejected for all of them, meaning that neither real data nor simulation data follow a normal distribution. To do a parametric correlation test such as Pearson correlation coefficient test, the data should be normalized [95]. Several techniques are available for normalizing datasets, from which three techniques of *standard* (A in Figure 10), *log transformation* (B in Figure 10), and *box-cox transformation* (C in Figure 10) were used. The normalization performance of these three techniques for aveWSN2g is shown in Figure 10. According to this figure, aveWSN2g data has not been normalized by any of these data transformation techniques. The results of the other fourteen blocks (each telecoms technology in each network structure) also indicate that the other datasets have not been made normalized, so a non-parametric technique has been used to measure the correlation coefficient of the model's outputs and historical real-world data.

**Table 11.** Results of Shapiro–Wilk test in 1–100 runs (block 1).

| Data | Description | Shapiro-Wilk Test | | |
|---|---|---|---|---|
| | | w | *p*-value | $H_0$ |
| Avepan2g | average simulated data of 2G generated in PAN | 0.92619 | 0.001374 | Rejected |
| Avepan3g | average simulated data of 3G generated in PAN | 0.94122 | 0.006143 | Rejected |
| Avepan4g | average simulated data of 4G generated in PAN | 0.93209 | 0.002438 | Rejected |
| Avewsn2g | average simulated data of 2G generated in WSN | 0.927 | 0.001485 | Rejected |
| Avewsn3g | average simulated data of 3G generated in WSN | 0.93371 | 0.002864 | Rejected |
| Avewsn4g | average simulated data of 4G generated in WSN | 0.92954 | 0.001898 | Rejected |
| R2g | Real world historical dataset of 2G diffusion in Iran | 0.88809 | 4.977e–05 | Rejected |
| R3g | Real world historical dataset of 3G diffusion in Iran | 0.94505 | 0.009171 | Rejected |
| R4g | Real world historical dataset of 4G diffusion in Iran | 0.90503 | 0.0002016 | Rejected |

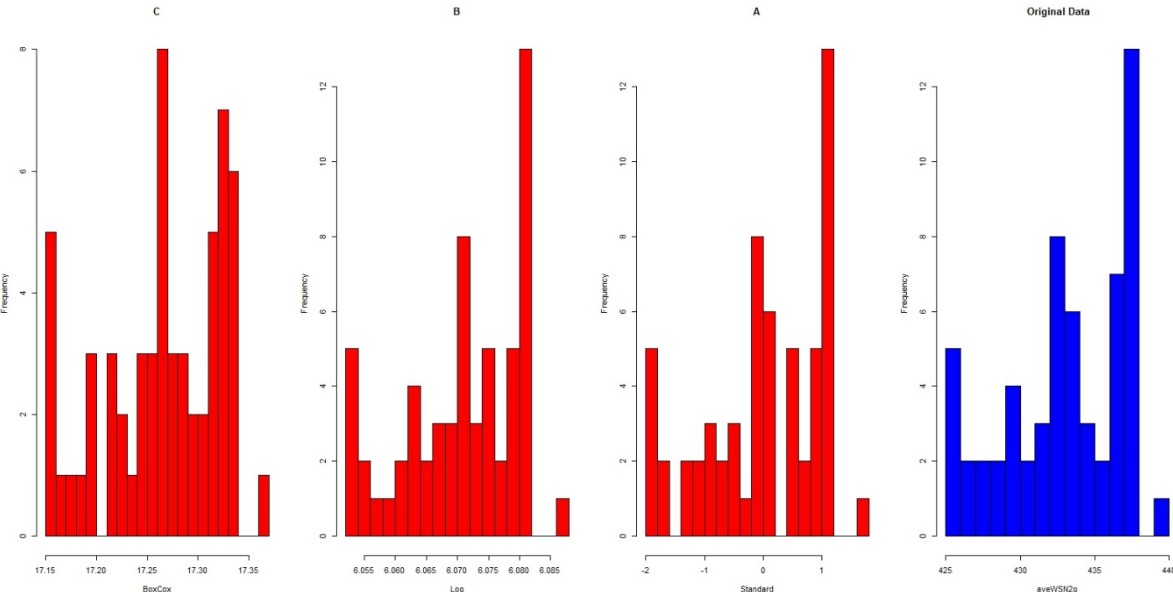

**Figure 10.** Normalization performance for aveWSN2g in 1–100 runs (block 1).

Correlation Coefficient Measurement

As a nonparametric method of correlation, the Spearman correlation coefficient test was used to measure the true correlation of simulated outputs data and real-world data. The results of block 1 are presented in Table 12.

**Table 12.** Spearman correlation coefficient Test results in 1–100 runs (block 1).

| Data | Description | Spearman Correlation Test | | | |
|------|-------------|-----|---------|-------------------|-------|
| | | S | *p*-value | Rho (correlation) | H$_0$ |
| Avepan2g | average simulated data of 2G generated in PAN | 147.02 | 2.2e–16 | 0.9959149 | Rejected |
| Avepan3g | average simulated data of 3G generated in PAN | 487.26 | <2.2e–16 | 0.9864614 | Rejected |
| Avepan4g | average simulated data of 4G generated in PAN | 111.87 | <2.2e–16 | 0.9968916 | Rejected |
| Avewsn2g | average simulated data of 2G generated in WSN | 157.55 | <2.2e–16 | 0.9970364 | Rejected |
| Avewsn3g | average simulated data of 3G generated in WSN | 510.99 | <2.2e–16 | 0.9858019 | Rejected |
| Avewsn4g | average simulated data of 4G generated in WSN | 111.87 | <2.2e–16 | 0.9968916 | Rejected |

H$_0$ Hypothesis signifies that the correlation of two samples (i.e., simulation data vs. real-world) is less than or equal to zero. As seen in column 3 of Table 12, all H$_0$ hypotheses are rejected (the p-value is far less than 0.05), meaning that the model has shown an acceptable predictive accuracy in both network structures. Therefore, the model's outputs are empirically validated. To pinpoint which social network has the highest performance and can serve as the best explanation for the social structure of the case study, the correlation coefficient of all blocks has been measured and presented in Table 13.

**Table 13.** Social networks performance in simulating the diffusion of mobile telecommunication technologies (2G, 3G, and 4G).

| Number of Blocks (Multiple Runs) | 2G | | 3G | | 4G | |
|---|---|---|---|---|---|---|
| | PAN | WSN | PAN | WSN | PAN | WSN |
| Block 1 (1–100) | 0.995914888 | 0.995622317 | 0.986461356 | 0.985801933 | 0.996891628 | 0.996891628 |
| Block 2 (101–200) | 0.995956402 | 0.996166451 | 0.982544504 | 0.985436519 | 0.996891628 | 0.996891628 |
| Block 3 (201–300) | 0.994838045 | 0.995984549 | 0.972365054 | 0.986351149 | 0.996891628 | 0.996891628 |
| Block 4 (301–400) | 0.995916081 | 0.996879707 | 0.985623227 | 0.9858684 | 0.996891628 | 0.996891628 |
| Block 5 (401–500) | 0.995132448 | 0.995636979 | 0.979471748 | 0.985073357 | 0.996891628 | 0.996891628 |
| Block 6 (501–600) | 0.994698311 | 0.995902483 | 0.985212857 | 0.985595798 | 0.996891628 | 0.996891628 |
| Block 7 (601–700) | 0.995230473 | 0.995425341 | 0.986106428 | 0.986400827 | 0.996891628 | 0.996891628 |
| Block 8 (701–800) | 0.99433489 | 0.994783757 | 0.982773917 | 0.985401794 | 0.996849922 | 0.996961433 |
| Block 9 (801–900) | 0.995719691 | 0.995470086 | 0.985447319 | 0.984708019 | 0.996891628 | 0.996891628 |
| Block 10 (901–1000) | 0.995928753 | 0.995132448 | 0.98628003 | 0.986012813 | 0.996891628 | 0.996891628 |
| Block 11 (1001–1100) | 0.995329409 | 0.994755081 | 0.981240056 | 0.986103608 | 0.996905604 | 0.996891628 |
| Block 12 (1101–1200) | 0.99492104 | 0.996601752 | 0.982942205 | 0.985013477 | 0.996891628 | 0.996891628 |
| Block 13 (1201–1300) | 0.996459663 | 0.996043596 | 0.977795559 | 0.984477854 | 0.996849922 | 0.996891628 |
| Block 14 (1301–1400) | 0.99482494 | 0.994937795 | 0.985328098 | 0.984968097 | 0.996891628 | 0.996891628 |
| Block 15 (1401–1500) | 0.995564687 | 0.994684451 | 0.986315611 | 0.98671933 | 0.996891628 | 0.996891628 |
| Total Mean | 0.995385 | 0.995601786 | 0.983060531 | 0.985595532 | 0.996886999 | 0.996896282 |

In terms of 2G simulation, as it can be seen in Figure 11, the performance of WSN (i.e., P$_{WSN}$) is better than that of PAN (i.e., P$_{PAN}$). The correlation performance of these two social networks in simulating the diffusion of 2G has been numerically presented in column 2 of Table 13.

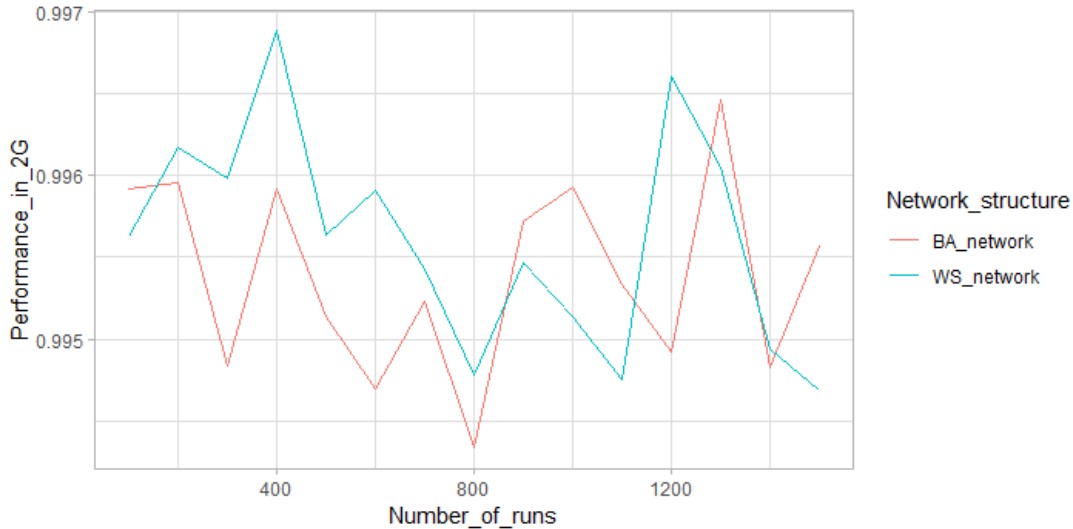

**Figure 11.** Performance of social structures in the simulation of 2G.

In terms of 3G simulation, $P_{WSN}$ is again better than $P_{PAN}$, as seen in Figure 12. All of these comparisons are presented in column 3 of Table 13.

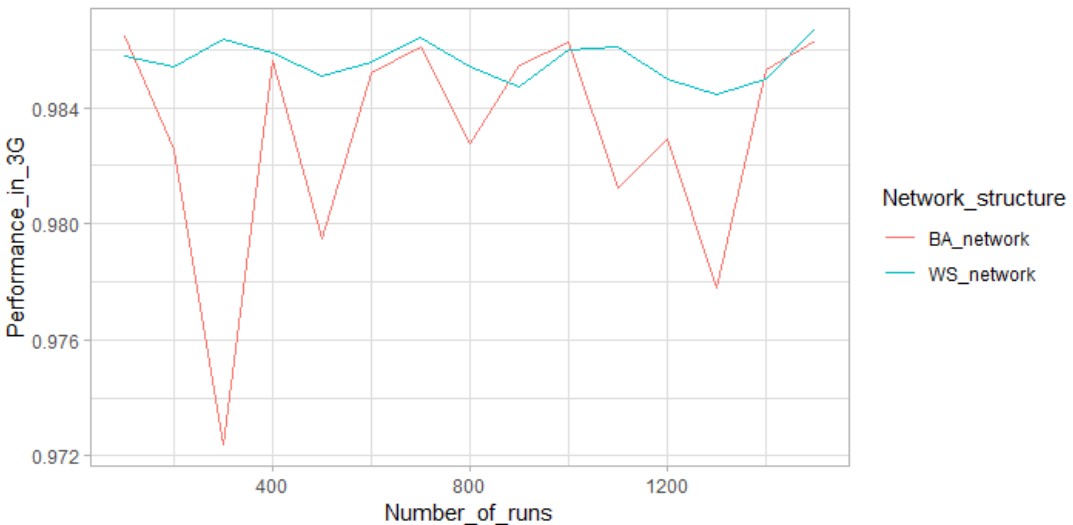

**Figure 12.** Performance of social structures in the simulation of 3G.

As can be seen in Figure 13 and column 4 of Table 13, the performance of WSN and PAN is a bit different in terms of 4G simulation, but WSN shows a better performance again. All results show that WSN has had a better performance in replicating the diffusion of MTTs in Iran; therefore, WSN seems to be the best explanation for the social structure of Iranian Society. A large number of studies support this finding that small-world networks are an efficient explanation for social networks [67,80,96–99] where the clustering coefficient (CC) is high (i.e., 0.34 for this study because of the size of sample, but it tends to $\frac{3}{4}$ (0.75) in case of large N) and average path length (L) is low (i.e., 2.90 for this study, but it will approximately converge to 7.90 for real number of Iranian population). High CC indicates that the friends of a person tend to be friends of each other, and low L shows that when two persons want to reach each other, they can do it by a limited number of steps. This network structure is highly efficient in information transfer where the world-of-mouth plays a critical role [100,101].

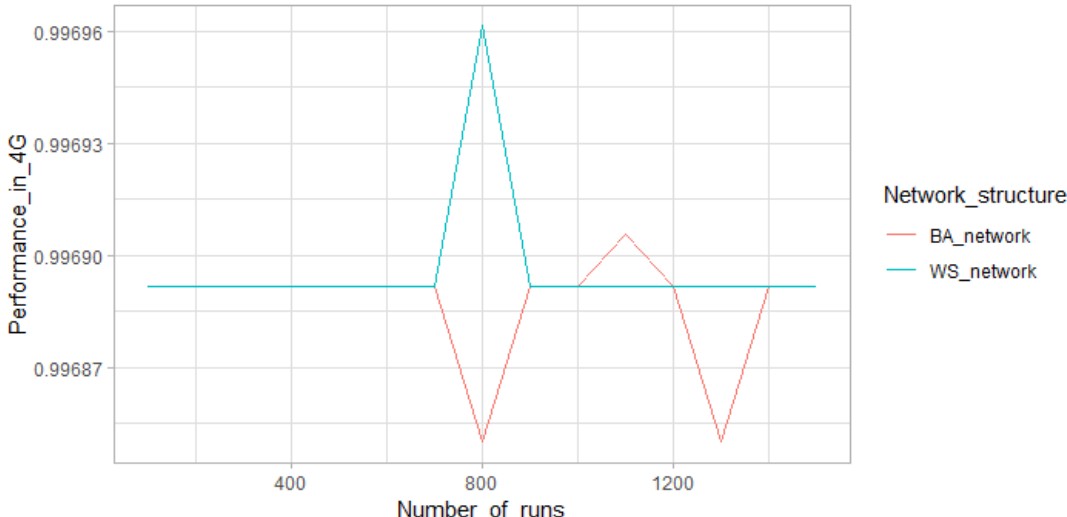

**Figure 13.** Performance of social structures in the simulation of 4G.

## 5. Discussion

### 5.1. Output Analysis

#### 5.1.1. Parameter Necessity Analysis

Now that the inputs and outputs have been empirically validated, the model can be used for scenario planning. The purpose of a scenario is to select a combination of operational steps, which increases technology diffusion by the highest possible degree. Operational steps are the product of decision parameters' states. The decision parameters are some of the input parameters that can be altered to monitor the targeted behavior of the system in a specific time domain. The decision parameters of this model are presented in Table 14. The model's outputs are validated against six months ranging from the beginning of summer 2017 to the end of autumn 2017 (see Appendix B), the validation results show that the model's predictive accuracy is very high for each output. Before scenario planning, two preliminary steps should be taken. The first step is to pinpoint on which parameter the diffusion of 4G is necessarily dependent. The second step is to identify the increase in which parameter has a higher effect on the increase of 4G diffusion.

**Table 14.** Decision parameters.

| Decision Parameter | Initial Value | Direction | Target | Time Steps |
|---|---|---|---|---|
| 4G-compatibility (4GComP) | 0.8 | Increase | 4G diffusion | 30 (each step for three days) |
| Advertisement (Ad) | 0.6 | Increase | 4G diffusion | 30 (each step for three days) |

All decision parameters have a continuous numerical scale (ratio) ranging from 0 to 1, as presented in Table 14. This scale is converted to a Boolean structure where the value of 1 stands for true (T) and 0 for false (F). The purpose of this conversion is to show without which parameter, the diffusion of 4G never happens (step 1). As shown in Figure 14, both compatibility and advertisement are tested for a value of 0. It shows that in case of compatibility = 0, the 4G has not changed in all 100 experiments, while the 4G has changed frequently in all 100 experiments (i.e., runs) in case of advertisement = 0. This result indicates that the compatibility parameter is necessary for the diffusion of 4G because the diffusion of 4G is impossible to happen without the existence of the compatibility parameter. Counter-intuitively, the 4G adoption is necessarily dependent on compatibility degree than the volume

of TV advertisement, which can be rooted in the technology push nature of the MTTs [102–104] and WSN topology of Iranian social network, where the world-of-mouth can drastically affect the information transfer regarding any artifact [100,101].

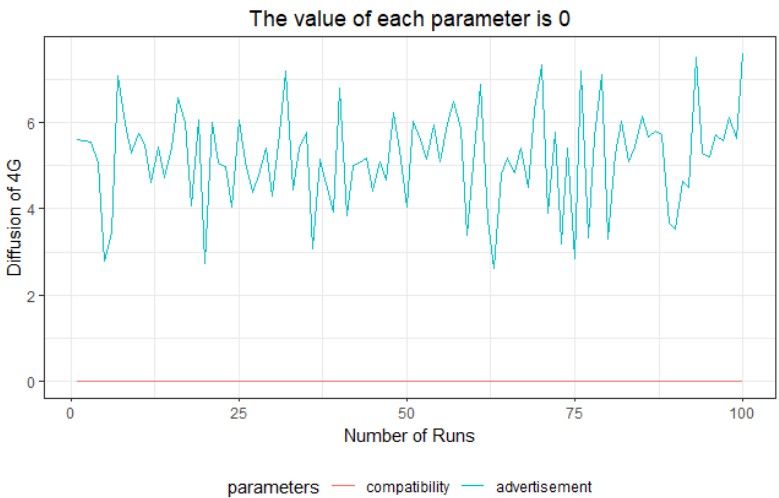

**Figure 14.** The impact of two parameters in case of value = 0.

### 5.1.2. Parameter Influence Analysis

For step 2, compatibility and advertisement are implemented for the value of 1, as shown in Figure 15. Though it shows the value of 4G diffusion has been changed by each of the parameters, in most of the experiments, the impact of compatibility has been much higher than that of advertisement in increasing the diffusion of 4G. All the results of steps 1 and 2 show that the compatibility parameter is not only necessary for 4G diffusion, but also has a much higher impact on 4G technology diffusion over time. Therefore, the compatibility value must never be zero if CT strategists are decided to scale up the diffusion of 4G technology.

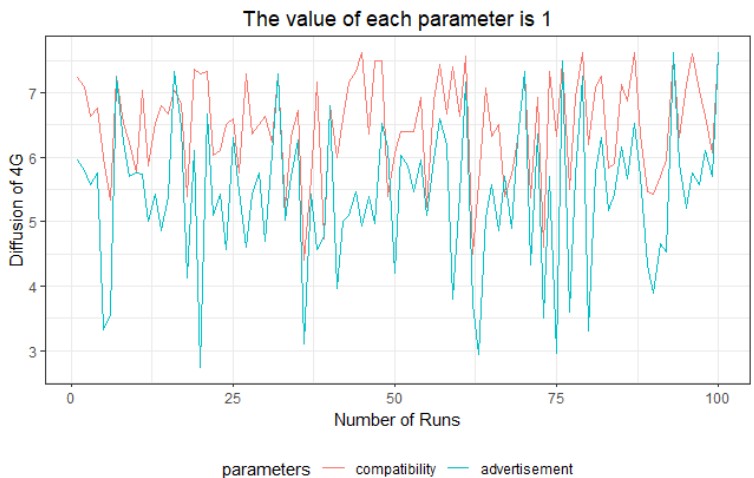

**Figure 15.** The impact of two parameters in case of value = 1.

### 5.1.3. Scenario Formulation

#### State-Step Matrix

For scenario formulation, all operational steps have to be specified. The number of all possible operational steps is the product of the number of states of each parameter. The states of compatibility parameter are [0.2,0.4,0.6,0.8,1], and the advertisement parameter states are [0, 0.2,0.4,0.6,0.8,1].

The compatibility parameter cannot take zero because the diffusion of technology is necessarily dependent upon it. Accordingly, all 30 operational steps (S) are presented in a state-step matrix of Table 15.

**Table 15.** State-Step matrix for scenario formulation.

| | | Advertisement | | | | | |
|---|---|---|---|---|---|---|---|
| | | **0** | **0.2** | **0.4** | **0.6** | **0.8** | **1** |
| **Compatibility** | 0.2 | $S_1$ (146.3240) | $S_2$ (146.3397) | $S_3$ (146.3437) | $S_4$ (146.3483) | $S_5$ (146.3627) | $S_6$ (146.3747) |
| | 0.4 | $S_7$ (147.7033) | $S_8$ (147.7220) | $S_9$ (147.7377) | $S_{10}$ (147.7590) | $S_{11}$ (147.7870) | $S_{12}$ (147.8180) |
| | 0.6 | $S_{13}$ (148.8360) | $S_{14}$ (148.8747) | $S_{15}$ (148.9187) | $S_{16}$ (148.9480) | $S_{17}$ (148.9950) | $S_{18}$ (149.0220) |
| | 0.8 | $S_{19}$ (150.1823) | $S_{20}$ (150.2427) | $S_{21}$ (150.2950) | $S_{22}$ (150.3437) | $S_{23}$ (150.3970) | $S_{24}$ (150.4277) |
| | 1 | $S_{25}$ (151.3257) | $S_{26}$ (151.3833) | $S_{27}$ (151.4510) | $S_{28}$ (151.5053) | $S_{29}$ (151.5647) | $S_{30}$ (151.6290) |

As presented in Table 15, the effect of each operational step S has been simulated by 100 experiment runs over a 3-month period (see Appendix C for details). The worst operational step is S1, where the average value of 4G diffusion has reached 146.3240, while the best operational step is S30, where the average value of 4G diffusion has reached 151.6290. The influence sequence of each operational step is shown in Figure 16.

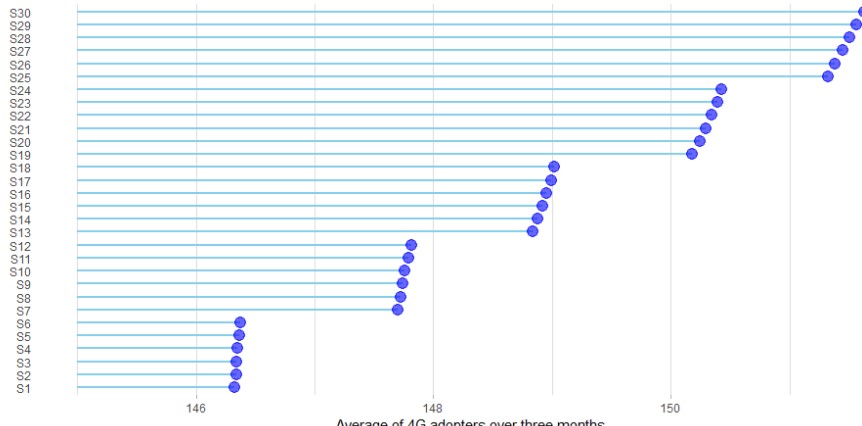

**Figure 16.** Influence the sequence of operational steps.

Scenario Extraction

A scenario is a sequence of operational steps with the purpose of increasing the diffusion of 4G technology by the highest possible degree over a specified time period. An appropriate scenario for such a purpose should have two necessary criteria. The first criterion is that it should have the least number of steps, and the second one is that its performance should show the least number of turning points. A large number of scenarios can be extracted from Table 15, but two of them only meet necessary criteria and can be considered appropriate. These scenarios are shown in Figures 17 and 18.

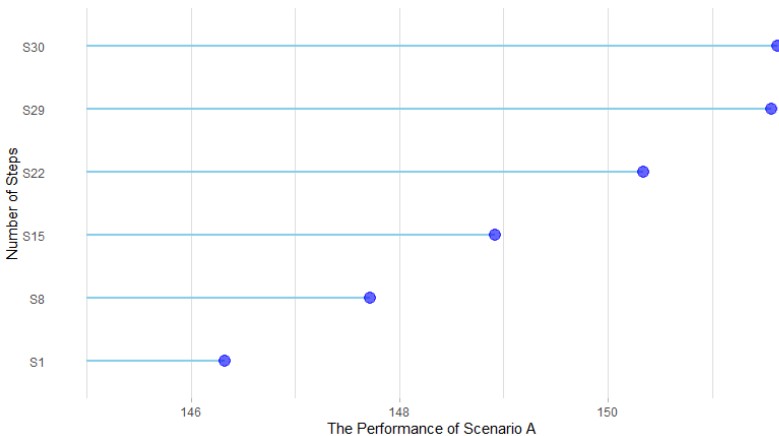

**Figure 17.** Scenario A.

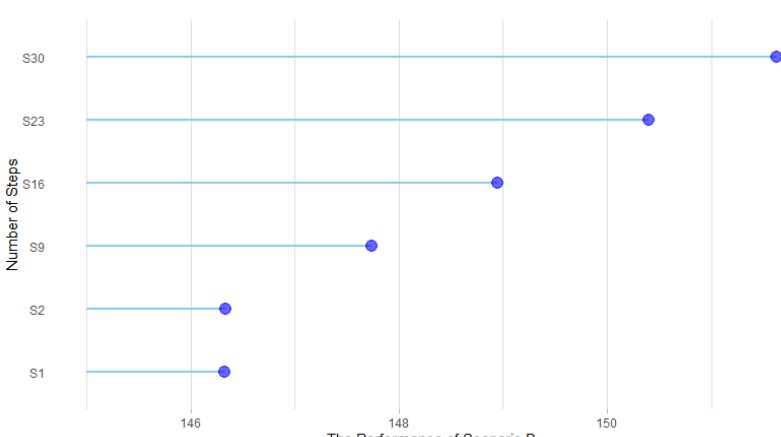

**Figure 18.** Scenario B.

## 6. Conclusions

Developing the appropriate scenarios for increasing the 4G technology diffusion in Iran has been the motivation behind the development of this model. In this regard, the mechanism underlying the diffusion of MTTs in Iranian society was studied. To better identify this mechanism, prior studies were first investigated, and because of reasons such as (1) simplistic neoclassical assumptions like unbounded rationality and perfect information, (2) non-inclusion of network effects and non-linear interactions of micro-level individual adopters, (3) learnability and adaptability of individual adopters, (4) bottom-up nature of technology adoption, (5) heterogeneity and multiplicity of influential factors, agent-based modeling (ABM) was chosen as the modeling approach in tandem with social network theory (SNT). This model is mainly theory-based, where real-world empirical data, theories, and extension of the works of [64–67] have been taken into account, and it is to a lesser extent evidence-based because some new variables (e.g., handset compatibility) are added authors.

The model's input parameters can be grouped as (1) number of adopters (i.e., 2G, 3G, and 4G), (2) adopters' memory parameters (i.e., innovators, early-adopters, early-majority, late-majority, and laggards), (3) network topology parameters (i.e., WSN and PAN) and (4) advertisement and handset compatibility parameters. They were directly calibrated according to empirical studies. The model was implemented by Netlogo 6.0.1, and the data were analyzed by R 3.5.3. The performance of the model was simulated in two different social network topologies by 9000 experiment runs for the second half of 2017 actually, 1500 simulation runs for each MTT in each social network structure. The Shapiro–Wilk test was used to check the normality of data, showing that none of them were normal. Data normalization methods such as standard (z-score), log transformation, and box-cox were used

to normalize the data, but none of them could normalize the data. Therefore, as a non-parametric test of correlation, spearman test was used to measure the model's outputs with real data for the second half of 2017, and the results indicated the high predictive accuracy of model's performance, and it was figured out that Watts–Strogatz small-world network (WSN) has the highest similarity with the social network of Iranian people where the clustering coefficient (CC) is high (i.e., 0.34 for this study because of the size of sample, but it tends to $\frac{3}{4}$ (0.75) in case of large N) and average path length (L) is low (i.e., 2.90 for this study, but it will approximately converge to 7.90 for real number of Iranian population). High CC indicates that the friends of a person tend to be friends of each other, and low L shows that when two persons want to reach each other, they can do it by a limited number of steps. Several number of studies affirmed this finding that social networks mainly follow a WSN structure [67,80,96–99].

In order to formulate scenarios for accelerating 4G diffusion, the decision parameters were selected from input parameters, and the necessity and influence of decision parameters were studied in 100 simulation runs for the first quarter of 2018. In terms of necessity, the results showed that if the value of advertisement is zero, 4G technology could get diffused over the society, which is mainly because of the technology push nature of MTTs [102–104] and WSN topology of Iran's social network, where world-of-mouth plays a critical role [100,101], while 4G could never be diffused if the handset compatibility is zero. In addition, in terms of influence, the compatibility parameter was found to have a higher performance in increasing 4G technology than advertisement parameter. Thereafter, a number of operational steps were extracted through the states of each parameter (i.e., six states for advertisement and five states for compatibility). Using these results, the most appropriate scenarios were extracted; scenarios that continuously increased 4G technology diffusion in Iran by the least number of operational steps and turning points. Like any scientific work, especially researches on empirical ABMs, this work has had some limitations. The first limitation was the inaccessibility and incompleteness of some individual-level data, such as the reluctance to innovation (R2I) score for which Individual Innovativeness (II) Scale was distributed among 1250 individuals, out of which 903 returned, and 800 of them were complete. The second limitation was the fact that some social recorded data were not up-to-date and needed to be collected from various sources.

Two topics can be recommended for future researches. The first topic is about the study of model application in other communities, which can provide useful information on model generalizability in other case studies. The second topic is that the entry of the fifth-generation of mobile technologies (5G) in 2020 [105,106] will definitely affect the diffusion of other MTTs, which investigating this effect is worth being the subject of a separate study.

**Author Contributions:** All authors have read and agree to the published version of the manuscript. Conceptualization, H.S. and M.A.S.; methodology, H.S.; software, H.S.; validation, H.S., M.A.S. and M.G.; formal analysis, H.S.; investigation, H.S.; data curation, H.S. and M.A.S.; writing—original draft preparation, H.S.; writing—review and editing, M.A.S.; visualization, H.S.; supervision, M.A.S.; project administration, M.G. and A.B.N.

**Funding:** This research received no external funding.

**Conflicts of Interest:** The authors declare no conflict of interest.

## Appendix A. Model User Interface in Netlogo 6.0.1

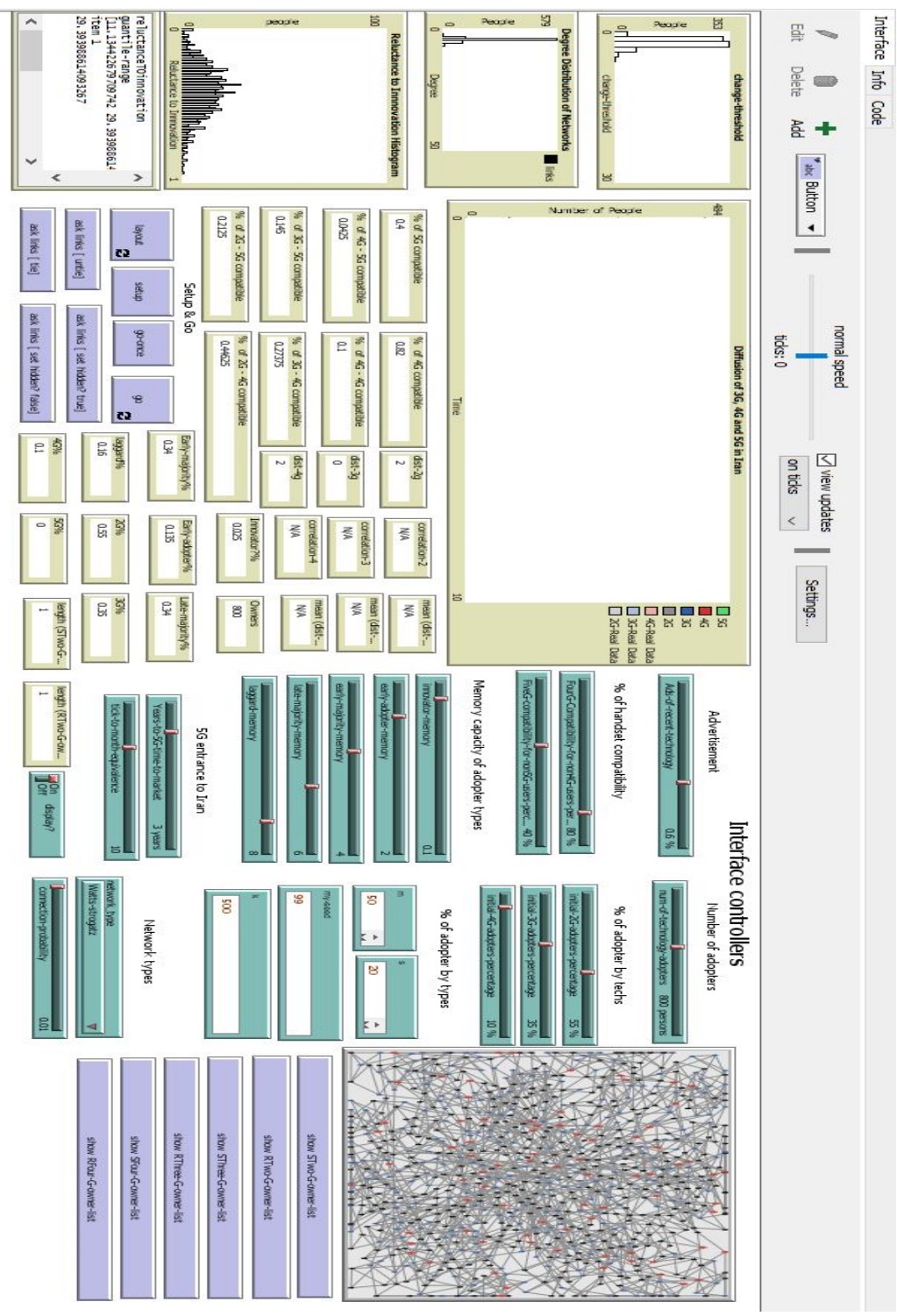

**Appendix B. The Real Data for the Diffusion of 2G, 3G, and 4G (from 7/1/2017 to 12/31/2017)**

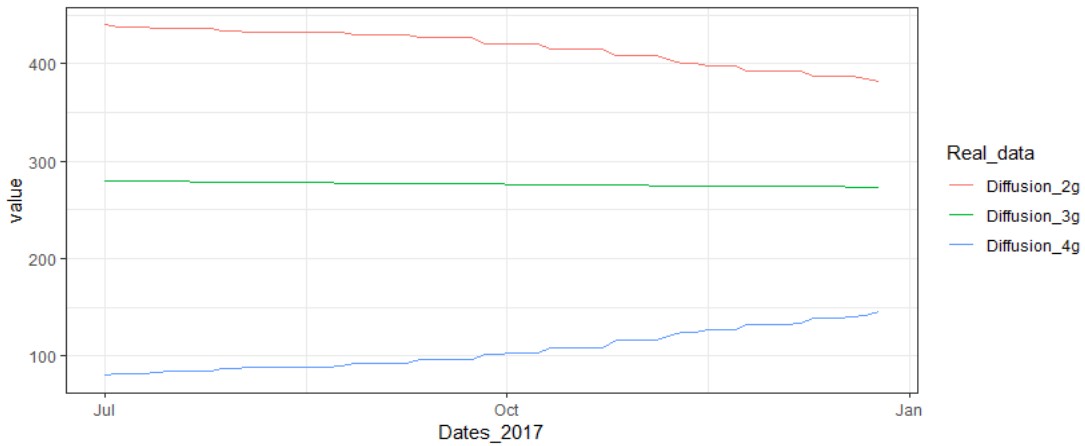

**Appendix C. The Average Effect of Each Scenario on Diffusion of LTE 4G over 100 Experiment Runs**

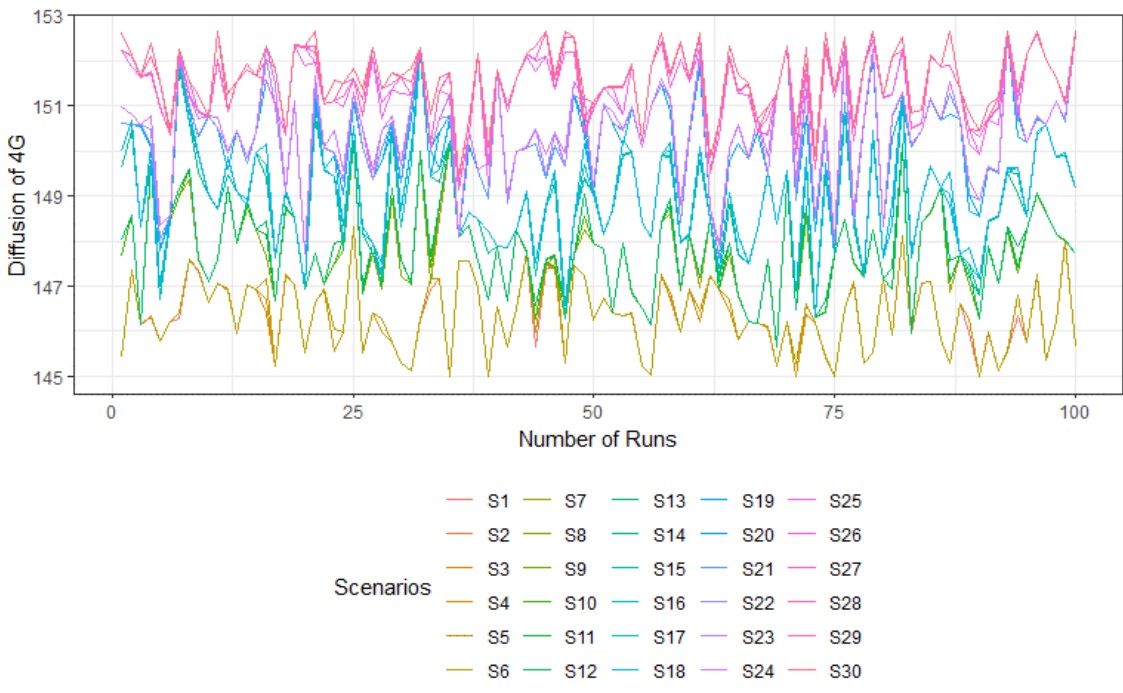

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
