# Peer review of "Modeling the Adoption and Diffusion of Mobile Telecommunications Technologies in Iran: A Computational Approach Based on Agent-Based Modeling and Social Network Theory"

_sustainability, doi:10.3390/su12072904_

Round 1
Reviewer 1 Report
The paper discusses the adoption of MTT generations, specifically 4G, in the Iranian context, and uses two social network theories as a basis for ABM-based analysis.
Some concerns:
First, the introduction should make it chrystal clear what data (collected from external reports) is concerns the past, and what concerns the present situation at a global and Iranian level and how it relates to future via the simulation. Now the text is partly vague in these terms. As Iran seems to come behind in 4G adoption, I would also advice the authors to check earlier research made or data published on the adoption of 4G in other countries. Ericsson Mobility reports are a very good reference, but there are several others available, too. This discussion on mobile technology adoption should also be reflected in the conclusions.
Second, the paper explains its approach and methodology in a very detailed way. Although this explanation has its obvious merits, I would recommend the authors to shorten the text to explain more how to the research is conducted than what the ABM as an approach is.
Third, I would like to see the results explained and opened up in the text a bit better.
Fourth, the conclusions could be improved in terms of what we can learn about this research regarding modeling MTT adoption, how the selected social network theories contribute to 4G adoption based on data from Iran and what are the limitations of the findings and data used. I would also like to see the key contribution and differentiation form earlier papers clearly stated.
Author Response
Dear Reviewer
It is my honor to receive your invaluable comments and I revised the paper accordingly.
- References of Ericsson mobility were added to the paper (for example[1] ).As you clearly pointed out, all data indicate the high deployment of 4G LTE by telecom operators among the countries. It also holds for Iran but the diffusion is still low in spite of high deployment. The Diffusion rate of 4G LTE was about 10% in July 2017 which became 14% in July 2019. I think It is majorly because of structural features of Iranian social network structure and the necessity of mobile handset compatibility that I have discussed them in the paper.
- The methodology part was reduced and the model implementation is discussed in a step-by-step sequence in order to increase the readability.
- We opened the results a bit more.
- We improved the conclusion more and the following points are discussed.
- The diffusion of MTTs should be investigated by a bottom-up approach which is rooted in the complexity theory. A theory that can yield a lots of realistic insights in contrast to reductionism school of thought( well discussed in [2, 3] as examples).
- Network theories points to the facts that the structure in which actors interact is extremely influential on what they act and react to. In terms of technology diffusion in a society, understanding the social structure can uncover the hidden drivers and barrier (which cannot be easily captured by other methodologies) as an instance, in a WS network, the power of the word-of-mouth outweighs a large array of other complex networks (It is well discussed in [4–7].
- Major differentiating points of paper were re-written such as 1) identification of Iranian Social network, 2) it was pointed out that the mobile handset compatibility parameter is counterintuitively far more critical and necessary that advertisement parameter, 3) extraction of strategies that can speed up 4G LTE adoption in Iran with regard to its social structure.
References
[1] Heuveldop, N. Ericsson Mobility Report June 2017. Ericsson, Tech. Rep., 2017.
[2] Epstein, J. M.; Axtell, R. Artificial Societies and Generative Social Science. Artif. Life Robot., 1997, 1 (1), 33–34.
[3] Elsenbroich, C.; Gilbert, N. Modelling Norms; Springer, 2014. https://doi.org/10.1007/978-94-007-7052-2.
[4] Wang, X. F.; Chen, G. Complex Networks: Small-World, Scale-Free and Beyond. IEEE circuits Syst. Mag., 2003, 3 (1), 6–20.
[5] Moore, C.; Newman, M. E. J. Epidemics and Percolation in Small-World Networks. Phys. Rev. E, 2000, 61 (5), 5678.
[6] Watts, D. J. A Simple Model of Fads and Cascading Failures. Prepr. available at< http//www. santafe. edu/sfi/publications/Abstracts/00-12-062abs. html, 2000.
[7] Moreno, Y.; Pastor-Satorras, R.; Vespignani, A. Epidemic Outbreaks in Complex Heterogeneous Networks. Eur. Phys. J. B-Condensed Matter Complex Syst., 2002, 26 (4), 521–529.
We thank you again for your precious comments and look forward to hearing from you.
Truly Yours
Authors
3/2/2020

Reviewer 2 Report
In this paper the authors propose an agent-based model intended to map the Iranian Mobile Telecommunication Technologies over time.
I suggest the authors to address the following problems/limitations:
In the abstract:
- Excessive use of acronyms. Mainly, only one acronym is repeated (MTT) in the abstract and thus, only this one is necessary.
- authors claim 3 contributions but none of them are clear. (1) seems to be an observation; (2) seems to be a finding of the model fit; (3) is a conclusion extrapolated for other studies. Authors should emphasize the accuracy of their model when compared to other ones and that this accuracy was assessed against real-world data.
In the intro section:
- good overview of the mobile technologies is provided. However, from line 45 to 91 there are no references provided, as many information, standards and corporations are cited.
- figures are in bitmap and not in vector format, this maybe a plus to correct as these graphs seem to be made with excel (when importing these to google sheets, the export to a vector format is straight forward)
- something seems to be missing before "6250" in line 95.
- introduction seems to be large and with many detailed information. The authors are suggest to address the main context, motivation and contributions in this section and insert detailed data in a state-of-the-art section, where it can be included a detailed context and information, and also the literature review.
In the literature review section:
- something seems to be missing in lines 224 and 225 after the commas and also before the "and" word.
- authors show discuss and draw a final conclusion about the literature review
In material and methods section:
- authors should redesign the flow chart in figure 5 in order to fit the page and improve readability.
- references are not found in text in lines 367, 369, 388, 396, 397, 411, 412, 449, 450, 477
- the acronym of "Watts-Strogatz small-world network" could be "WS network", and "preferential attachment network" could be "PA network" in order to avoid confusion with other expansions possible.
- figure 7 is in vector format which is valuable but it is used a small zoom that turns the image to be unreadable. Authors should redraw to fit in a vertical page and improve readability
In results section:
- references are not found in text in lines 528, 547, 548, 683, 694, 724, 728, 744
- figure 8 is in vector format which is valuable but it is used a small zoom that turns the image to be unreadable.
- excessive structure division is implemented. I this type of articles a level 4 section could be admitted in some special descriptions, however authors not only use level 4 but also level 5 as in "4.1.2.1.1". This turns the document to be more complex and less easy to understand and follow.
- figure 9, 10, 11, 12, 14, 15, and 16 are not in vector format and could be improved
- table 15 is not presented adequately
In conclusions section:
- the section could be changed to allow better readability. The description is extensive and should be improved.
In appendix:
- images could present more detail or resolution
As an overall consideration, the authors should improve all the paper readability, should simplify descriptions and highlight their contributions. Also, they should present validations and should better explain all their work. Some of the parts of this document are clearly not easy to follow.
Author Response
Dear Reviewer
It is my honor to receive your invaluable comments and I revised the paper accordingly.
For abstract
- Acronyms were reduced only to MMTs
- The paper contribution is better explained and the empirical validation of paper is discussed.
For introduction
- Some references were added to lines 45 to 91.
- According to authors guidelines of the Journal, formats such as JPEG are preferred but to increase the resolution in the level of a vector format, we increase all pictures resolution by DPI (dots per inch) of 600 using https://clideo.com/dpi-converter. It should be noted that Figures 1 and 2 were completely redrawn. Moreover, a file of all figures is attached to the paper as a Figures File.
- Line 95 was edited and a backspace was deleted before “6250”.
For literature Review
- Backspaces in lines 224 and 225 were deleted.
- A conclusion is drawn for this part.
For Materials and Methods
- Figure 5 was redesigned and its DPI increased to 600
- Some references were added for lines 367, 369, 388, 396, 397, 411, 412, 449, 450, 477.
- Acronyms for Watts-Strogatz small-world network (WS Network) and Barabási–Albert preferential attachment network (PA network) have a bit change.
- Figure 7 was completely redrawn for fitting in page.
For Results part
- Some references were added to lines 528, 547, 548, 683, 649, 724, 728, 744.
- Figure 8 was improved and its readability increased.
- structure divisions were reduced.
- Figures 9,10,11,12, 14, 45 and 16 were improved for readability. They all can also be found in the submitted Figures File.
- Table 15 is adequately presented through a little change.
For conclusion part
- Its readability is increased and the contributions are more clearly defined.
For appendix
- All Figures are improved through increasing DPI by 600 which all can also be found in the submitted Figures File.
We thank you again for your precious comments and look forward to hearing from you.
Truly Yours
Authors
3/2/2020

Round 2
Reviewer 1 Report
My concerns presented in the first round review have been answered. There are some typos in the text that could be corrected. Otherwise ok.